# Bi-Level Decision-Focused Causal Learning for Large-Scale Marketing Optimization: Bridging Observational and Experimental Data

**Shuli Zhang** [1*†]   **Hao Zhou**[1,2*‡]   **Jiaqi Zheng**[1‡]   **Guibin Jiang**[2]
**Bing Cheng**[2]   **Wei Lin**[2]   **Guihai Chen**[1]

[1]State Key Laboratory for Novel Software Technology, Nanjing University, Nanjing, China
[2]Meituan, Beijing, China
zhangshuli@smail.nju.edu.cn  {jzheng, gchen}@nju.edu.cn
{zhouhao29, jiangguibin, bing.cheng, linwei31}@meituan.com

## Abstract

Online Internet platforms require sophisticated marketing strategies to optimize user retention and platform revenue — a classical resource allocation problem. Traditional solutions adopt a two-stage pipeline: machine learning (ML) for predicting individual treatment effects to marketing actions, followed by operations research (OR) optimization for decision-making. This paradigm presents two fundamental technical challenges. First, the prediction-decision misalignment: Conventional ML methods focus solely on prediction accuracy without considering downstream optimization objectives, leading to improved predictive metrics that fail to translate to better decisions. Second, the bias-variance dilemma: Observational data suffers from multiple biases (e.g., selection bias, position bias), while experimental data (e.g., randomized controlled trials), though unbiased, is typically scarce and costly — resulting in high-variance estimates. We propose **Bi**-level **D**ecision-**F**ocused **C**ausal **L**earning (**Bi-DFCL**) that systematically addresses these challenges. First, we develop an unbiased estimator of OR decision quality using experimental data, which guides ML model training through surrogate loss functions that bridge discrete optimization gradients. Second, we establish a bi-level optimization framework that jointly leverages observational and experimental data, solved via implicit differentiation. This novel formulation enables our unbiased OR estimator to correct learning directions from biased observational data, achieving optimal bias-variance tradeoff. Extensive evaluations on public benchmarks, industrial marketing datasets, and large-scale online A/B tests demonstrate the effectiveness of Bi-DFCL, showing statistically significant improvements over state-of-the-art. Currently, Bi-DFCL has been deployed across several marketing scenarios at Meituan, one of the largest online food delivery platforms in the world.

## 1   Introduction

Marketing is one of the most effective strategies for enhancing user engagement and platform revenue, and as such, a variety of marketing campaigns have been widely adopted by online platforms. For instance, coupons on Taobao[45] stimulate user activity, dynamic pricing on Airbnb[44] and discounts on Uber[13] encourage increased usage. However, while these actions can generate incremental

---

[*]Both authors contributed equally to this research.

[†]Work was done during an internship at Meituan.

[‡]Corresponding author.

39th Conference on Neural Information Processing Systems (NeurIPS 2025).

revenue, they also consume substantial marketing resources such as budget. Due to these constraints, only a subset of individuals (e.g., shops or products) can receive marketing treatments. Therefore, determining how to allocate marketing resources effectively—given that users respond differently to various promotional offers—is crucial for campaign success. This challenge is typically formulated as a resource allocation problems, which has been extensively studied in both academia and industry.

The mainstream solutions to these problems are two-stage methods (TSM) [2, 48, 3, 39, 13]. In the first stage (ML), machine learning models are used to predict individual-level (incremental) responses to different treatments. In the second stage (OR), these predictions are fed into combinatorial optimization algorithms to maximize overall revenue. Hence, existing two-stage methods focus on separately optimizing the prediction and the subsequent resource allocation, treating them as decoupled problems. Despite widespread use, TSM suffer from two fundamental technical challenges:

- **The prediction-decision misalignment:** ML focuses on predictive accuracy, while OR aims for decision quality. However, improved prediction accuracy does not necessarily yield better decisions in many predict-then-optimize scenarios [23, 14, 32], due to the decoupled design. This misalignment is especially pronounced in marketing for two reasons. First, marketing OR problems are typically non-convex and NP-hard resource allocation tasks, which can amplify or accumulate prediction errors from the ML stage when passed to OR. Second, marketing involves counterfactual challenges (discussed later), making accurate predictions even harder. As a result, two-stage methods often lead to suboptimal decisions in marketing optimization.

- **The bias-variance dilemma:** In marketing optimization and causal inference [31], observational (OBS) data are abundant and easy to collect, e.g., from user behavior logs or transactions. However, such data are inherently biased due to confounding and lack of randomization, leading to **high bias and low variance**. In contrast, randomized controlled trials (RCTs) [10, 36] are considered the gold standard for causal inference, as randomization provides experimental data that yield unbiased estimates. Yet, RCT data are costly and limited in size, resulting in **low bias and high variance**, which also increases the risk of overfitting and reduces generalization. While OBS and RCT data are complementary, two-stage methods that rely solely on one type or naively combine both fail to achieve an effective bias-variance tradeoff, limiting robust decision-making in marketing.

Recently, Decision-Focused Learning (DFL) [23, 29, 4, 27, 14] has emerged as a promising alternative to traditional TSM by integrating ML and OR objectives within an end-to-end framework, specifically designed to address the Prediction-decision misalignment. The core idea is to train ML models using a loss function that directly reflects the quality of the resulting decisions. However, applying general DFL methods to marketing optimization raises unique challenges, including complexity of multi-choice knapsack problem (MCKP), constraint uncertainty, counterfactuals, computational cost of large-scale marketing data [52]. To tackle these domain-specific issues of marketing optimization, two specialized DFL approaches—DHCL [51] and DFCL [52]—have been proposed for marketing scenarios. While DHCL and DFCL have made notable progress in narrowing the gap between prediction and decision objectives (Challenge 1 of TSM), they do not fully resolve this misalignment, and further improvements are needed. Moreover, these methods may even exacerbate the bias-variance dilemma (Challenge 2 of TSM), as will be discussed in detail in Sec. 2.

In this work, we propose **Bi**-Level **D**ecision-**F**ocused **C**ausal **L**earning (**Bi-DFCL**). The key idea is to establish a bi-level optimization framework that leverages RCT data to end-to-end train an auxiliary Bridge Network by minimizing our proposed unbiased OR estimator, which in turn dynamically corrects the training direction on OBS data. By bridging OBS and RCT data, this design enables Target Network to better capture unbiased task-specific knowledge and address both the prediction-decision misalignment and bias-variance dilemma in TSM and DFL. We summarize our main contributions as:

- **Bridging the prediction-decision gap:** We propose an unbiased estimator of decision quality within the DFL paradigm and design two innovative surrogate decision losses leveraging RCT data. Such losses enable exact and efficient gradient computation for discrete optimization and, by operating on the primal problem, directly target the actual budget constraints of real-world marketing—leading to a more practical and consistent alignment between prediction and decision.

- **Addressing the bias-variance dilemma:** We establish a bi-level optimization framework that bridges OBS and RCT data. This architecture enables our unbiased OR estimator to dynamically correct the learning direction from biased OBS data via an auxiliary Bridge Network, achieving

optimal bias-variance trade-off. We further develop an implicit differentiation-based algorithm for bi-level optimization, ensuring end-to-end differentiability and scalability for large-scale marketing.

- **Adaptive multi-objective loss balancing:** By explicitly assigning prediction and decision losses the lower and upper levels of bi-level optimization, Bi-DFCL automatically and flexibly balances these objectives in a data-driven manner, eliminating the need for manual hyperparameter tuning.

- **Comprehensive offline and online validation:** We conduct extensive offline experiments on public benchmarks and industrial marketing datasets, as well as large-scale online A/B tests at Meituan, one of the largest online food delivery platforms in the world. Results show that Bi-DFCL consistently outperforms state-of-the-art methods. Notably, Bi-DFCL has already been deployed in several real-world marketing scenarios on this platform, generating significant revenue gains.

## 2 Related Works

**Two-Stage Method (TSM).** The mainstream approach to the resource allocation problem in marketing typically adopts a two-stage paradigm [3, 39, 48, 2, 13], in which the machine learning (ML) and operations research (OR) stages are addressed independently. In the first stage, uplift models are employed to predict the individual treatment effects. In the second stage, the resource allocation task is formulated as a multi-choice knapsack problem (MCKP), which is NP-hard but can be efficiently solved using Lagrangian duality theory [2, 3, 39, 51]. Note that the core idea of these methods is to continuously improve the predictive accuracy of the uplift models in the first stage. Accordingly, prior studies have focused on the design of uplift models, which can be categorized into four main groups: meta-learners [17, 24], causal forests [5, 38, 49, 2], reweighting-based methods [48, 40, 41, 9, 47, 18], and representation learning approaches [16, 43, 35, 8, 19]. However, as discussed in Sec. 1, TSM suffers from misalignment between prediction and decision objectives and fails to achieve an effective bias-variance tradeoff. Thus, even with improved predictive accuracy from advanced uplift models, better predictive metrics often do not translate into better or more robust decision quality.

**Decision-Focused Learning (DFL).** DFL offers an appealing alternative to the traditional two-stage approach by integrating prediction and optimization into an end-to-end framework. However, computing the decision loss typically involves solving optimization problems with non-differentiable operations, making it difficult for automatic differentiation tools in machine learning frameworks such as PyTorch [26] and TensorFlow [1] to provide correct gradients. Prior work has proposed three main strategies for gradient computation: (1) differentiating optimality conditions (e.g., via KKT or self-dual formulations, as in OptNet [4], DQP [12], QPTL [42], and IntOpt [21]), (2) smoothing by random perturbations and treating the optimization as a black box (e.g., DBB [27], DPO [7], I-MLE [25]), and (3) using surrogate loss functions (e.g., SPO [14], LTR [22], LODL [32], TaskMet[6], Lancer[50]). The first approach is limited to convex quadratic or linear programs, which do not fit settings of resource allocation problems. The second, while more general, is computationally expensive and impractical for large-scale marketing data. The third relies on access to optimal solutions, which are typically unobservable in offline marketing scenarios due to counterfactuals. As a result, effectively applying DFL to real-world marketing resource allocation remains challenging.

We emphasize that although existing DFL methods can address the inconsistency between prediction and decision objectives, none can be directly applied to marketing optimization due to domain-specific challenges such as the multi-choice knapsack problem, constraint uncertainty, counterfactuals, and the computational demands of large-scale datasets. Therefore, the most relevant works to ours are two DFL applications in marketing: DHCL [51] and DFCL [52]. DHCL directly learns an unbiased estimator of the decision factor in OR by customized loss, while DFCL introduces two surrogate losses (DFCL-DPL and DFCL-DIFD) for effective gradient estimation of the dual decision loss within the DFL paradigm. However, both approaches still have two notable limitations:

- **Exacerbation of the bias-variance dilemma.** In DHCL and DFCL, counterfactuals prevent direct computation of decision loss, so it can only be unbiasedly estimated from RCT data. Thus, abundant OBS data cannot be used for training, and learning is limited to scarce RCT samples, making models prone to overfitting and poor generalization (low bias but high variance).

- **Insufficiency in addressing prediction-decision misalignment.** DFCL still faces two key issues in aligning prediction and decision objectives. First, its loss is a weighted sum of decision and prediction losses, with the trade-off controlled by a manually tuned hyperparameter $\alpha$, which is inflexible and not fully automated. Second, DFCL uses a dual decision loss that evaluates quality

across all possible budgets, while real-world marketing budgets are typically limited to a narrow or discrete set. This mismatch can reduce alignment with actual decision quality in practice.

## 3 Problem Formulation

We initiate our formal analysis with a marketing optimization scenario involving $M$ distinct treatments. For each individual-treatment pair $(i, j)$, let $r_{ij} \in \mathbb{R}^+$ and $c_{ij} \in \mathbb{R}^+$ denote the potential revenue and associated cost respectively. The constrained optimization objective requires developing an allocation policy $\pi : [N] \to [M]$ that maximizes the platform's cumulative revenue under a global budget constraint $B$. This combinatorial decision-making challenge, which we term the Multi-Treatment Budget Allocation Problem (MTBAP), admits the following primal and dual formulations:

$$
\begin{aligned}
\max_z \quad & H(z; r, c) = \sum_i \sum_j z_{ij} r_{ij} \\
\text{s.t.} \quad & \sum_i \sum_j z_{ij} c_{ij} \le B \\
& \sum_j z_{ij} = 1, \, \forall i \in [N] \\
& z_{ij} \in \{0, 1\}, \, \forall i \in [N], \, j \in [M]
\end{aligned}
\qquad
\min_{\lambda \ge 0}
\left\{
\begin{aligned}
\max_z \quad & \left[ \lambda B + \sum_i \sum_j (r_{ij} - \lambda c_{ij}) z_{ij} \right] \\
\text{s.t.} \quad & \sum_j z_{ij} = 1, \, \forall i \in [N] \\
& z_{ij} \in \{0, 1\}, \, \forall i \in [N], \, j \in [M]
\end{aligned}
\right\}
$$

Figure 1: The primal (left) and dual (right) formulations of the MTBAP.

The binary variable $z_{ij} \in \{0, 1\}$ indicates whether individual $i$ is assigned treatment $j$. The primal problem is an instance of the NP-Hard MCKP [37]. The Lagrangian relaxation algorithm $\mathcal{A}$ (see Appendix A.1) efficiently finds the optimal solution to dual problem via binary search for $\lambda^*$, yielding an approximate solution to primal problem with a worst-case approximation ratio of $\rho = 1 - \frac{\max_{ij} r_{ij}}{\text{OPT}}$:

$$
z_{ij}^* = \mathcal{A}(H(z; r, c)) = \mathbb{1} \left\{ j = \arg \max_{j' \in [M]} \left[ r_{ij'} - \lambda^* c_{ij'} \right] \right\}. \tag{1}
$$

where $\mathbb{1}$ is indicator function. Let $\theta$ denote the parameters of Target Network $\mathcal{F}_\theta$, with $\hat{r}(\theta)$ and $\hat{c}(\theta)$ representing the predicted revenue and cost for individuals under different treatments, respectively. The prediction loss $\mathcal{L}_{\text{PL}}(\theta)$ is defined as the following MSE Loss between predicted and true values:

$$
\mathcal{L}_{\text{PL}}(\theta) = \mathbb{E}_{i \in [N], \, j \in [M]} \left[ (r_{ij} - \hat{r}_{ij}(\theta))^2 + (c_{ij} - \hat{c}_{ij}(\theta))^2 \right] \tag{2}
$$

Given predicted parameters $\hat{r}(\theta)$ and $\hat{c}(\theta)$, the allocation policy $z^*(\hat{r}(\theta), \hat{c}(\theta))$ is obtained by applying algorithm $\mathcal{A}$ to the optimization problem $H(z; \hat{r}(\theta), \hat{c}(\theta))$, as shown in eq.1 and Appendix A.1. The decision loss $\mathcal{L}_{\text{DL}}$ directly quantifies decision quality through the negative realized objective value:

$$
\mathcal{L}_{\text{DL}}(\theta) = -M \cdot \mathbb{E}_{i \in [N], \, j \in [M]} \left[ z_{ij}^*(\hat{r}(\theta), \hat{c}(\theta)) \cdot r_{ij} \right] \tag{3}
$$

Note that the prediction loss $\mathcal{L}_{\text{PL}}$ enhances model generalizability by minimizing estimation errors, whereas the decision loss $\mathcal{L}_{\text{DL}}$ evaluates policy suboptimality in downstream OR tasks and enables real-time decision quality awareness of the model. Thus, the composite objective $\mathcal{L}_{\text{DFCL}}$ in DFCL[52] is formulated to explicitly captures the dual objectives of predictive accuracy and decision quality as:

$$
\mathcal{L}_{\text{DFCL}} = \mathcal{L}_{\text{DL}} + \alpha \mathcal{L}_{\text{PL}} \tag{4}
$$

In digital marketing causal inference, each sample is represented by $(X, T, R, C)$, where $x_i$ denotes user features, $t_i$ the assigned treatment index, and $(r_{it_i}, c_{it_i})$ the observed factual revenue-cost pair under Rubin's potential outcomes framework [31]. The complete counterfactual surfaces $(R(t), C(t))$ remain partially observable across two distinct data modalities: experimental data $\mathcal{D}_{\text{RCT}}$ from randomized controlled trials satisfies strong ignorability $(X, R(t), C(t)) \perp T$ yet suffers from prohibitive collection costs and scarcity, whereas observational data $\mathcal{D}_{\text{OBS}}$ provides abundant samples via passive collection at the expense of confounding biases due to non-random treatment assignment.

The fundamental challenge in causal inference originates from Rubin's *missing counterfactual problem*: for any individual $i$ exposed to treatment $t_i$, only the factual outcome $(r_{it_i}, c_{it_i})$ is observed, while the counterfactual responses $\{(r_{ij}, c_{ij})\}_{j \ne t_i}$ remain fundamentally unobserved. This inherent data incompleteness implies the ground-truth values $\{r_{ij}, c_{ij}\}_{j=1}^M$ can never be fully ascertained, making both prediction loss $\mathcal{L}_{\text{PL}}$ and decision loss $\mathcal{L}_{\text{DL}}$ non-computable given either $\mathcal{D}_{\text{RCT}}$ or $\mathcal{D}_{\text{OBS}}$.

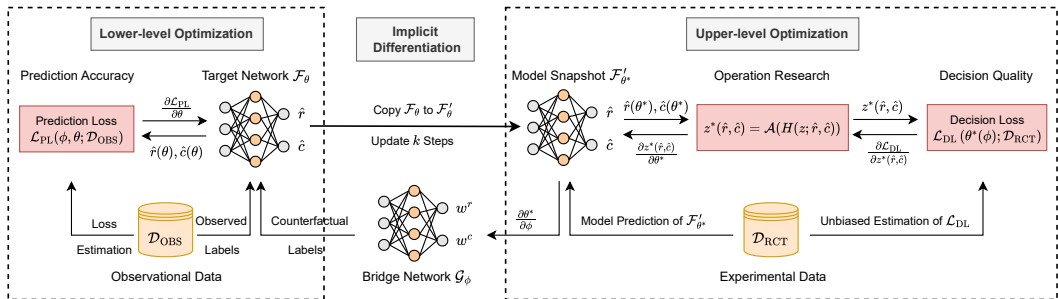

Figure 2: Overview of the Bi-Level Decision-Focused Causal Learning (Bi-DFCL) Framework.

# 4 Proposed Methods

## 4.1 Bi-level Optimization Framework

As discussed in Sec. 3, our objective is to minimize the composite loss $\mathcal{L}_{\text{DFCL}} = \mathcal{L}_{\text{DL}} + \alpha\mathcal{L}_{\text{PL}}$. While the DFCL framework [52] minimizes this loss solely on $\mathcal{D}_{\text{RCT}}$ to ensure unbiasedness, this approach overlooks complementary strengths of both $\mathcal{D}_{\text{RCT}}$ and $\mathcal{D}_{\text{OBS}}$, as well as distinct advantages of $\mathcal{L}_{DL}$ and $\mathcal{L}_{PL}$. Specifically, $\mathcal{L}_{\text{DL}}$ is highly dependent on the unbiasedness of $\mathcal{D}_{\text{RCT}}$: minimizing $\mathcal{L}_{\text{DL}}$ on biased $\mathcal{D}_{\text{OBS}}$ would greatly amplify bias and severely degrade decision quality. Conversely, $\mathcal{L}_{\text{PL}}$ is designed to improve generalization and is most effective when optimized on low-variance, large-scale $\mathcal{D}_{\text{OBS}}$. Ultimately, our goal is for a Target Network $\mathcal{F}_\theta$ trained on $\mathcal{D}_{\text{OBS}}$ to achieve high decision quality on $\mathcal{D}_{\text{RCT}}$. Motivated by this, we propose applying $\mathcal{L}_{\text{DL}}$ to $\mathcal{D}_{\text{RCT}}$ and $\mathcal{L}_{\text{PL}}$ to $\mathcal{D}_{\text{OBS}}$, assigning them to the upper and lower levels of a bi-level optimization framework, respectively.

$$\phi^* = \arg\min_\phi \mathcal{L}_{\text{DL}}\left(\theta^*(\phi); \mathcal{D}_{\text{RCT}}\right) \tag{5}$$

$$\text{s.t. } \theta^*(\phi) = \arg\min_\theta \mathcal{L}_{\text{PL}}(\phi, \theta; \mathcal{D}_{\text{OBS}}). \tag{6}$$

$\theta$ and $\phi$ denote parameters of Target Network $\mathcal{F}_\theta$ and Bridge Network $\mathcal{G}_\phi$, respectively. This setup constitutes a bi-level optimization (BLO) problem [46, 15, 28, 20], where the upper-level (5) and the lower-level (6) are nested: the objective and variables of upper level depend on the optimizer of lower level. The core idea is to end-to-end learn $\mathcal{G}_\phi$ by minimizing $\mathcal{L}_{\text{DL}}$ on $\mathcal{D}_{\text{RCT}}$, such that the parameterized prediction loss $\mathcal{L}_{\text{PL}}(\phi, \theta)$ on $\mathcal{D}_{\text{OBS}}$ is adaptively refined. The bridge vector $w$, output by $\mathcal{G}_\phi$, is dynamically updated and used to generate counterfactual pseudo-labels $r_{i,j}^{\text{cf}}, c_{i,j}^{\text{cf}}$ on $\mathcal{D}_{\text{OBS}}$:

$$r_{i,j}^{\text{cf}} = \hat{r}_{i,j}^{\text{pre}}(\psi) \cdot w_{i,j}^r + \hat{r}_{i,j}(\theta) \cdot (1 - w_{i,j}^r), \quad c_{i,j}^{\text{cf}} = \hat{c}_{i,j}^{\text{pre}}(\psi) \cdot w_{i,j}^c + \hat{c}_{i,j}(\theta) \cdot (1 - w_{i,j}^c), \tag{7}$$

$$w_{i,j}^r = \text{sigmoid}(\mathcal{G}_\phi^r(i, j)), \quad w_{i,j}^c = \text{sigmoid}(\mathcal{G}_\phi^c(i, j)) \tag{8}$$

Here, $w$ acts as a gating coefficient, adaptively combining outputs from $\mathcal{F}_\theta$ and a fixed teacher Network $\mathcal{F}_\psi$ (pretrained on $\mathcal{D}_{\text{RCT}}$ via any uplift model and kept fixed). This mechanism bridges $\mathcal{D}_{\text{OBS}}$ and $\mathcal{D}_{\text{RCT}}$ and generates stable counterfactual pseudo-labels to parameterize $\mathcal{L}_{\text{PL}}$ on $\mathcal{D}_{\text{OBS}}$:

$$\mathcal{L}_{\text{PL}}(\phi, \theta) = \mathbb{E}_{i,t_i}\left[(r_{it_i} - \hat{r}_{it_i})^2 + (c_{it_i} - \hat{c}_{it_i})^2\right] + \mathbb{E}_{i,j\neq t_i}\left[(r_{i,j}^{\text{cf}} - \hat{r}_{ij})^2 + (c_{i,j}^{\text{cf}} - \hat{c}_{ij})^2\right]. \tag{9}$$

By fully leveraging unbiased decision signals from $\mathcal{D}_{\text{RCT}}$, this approach makes the lower-level (6) both decision-aware and less biased, dynamically correcting the learning direction of Target Network $\mathcal{F}_\theta$. Assigning $\mathcal{L}_{\text{DL}}$ and $\mathcal{L}_{\text{PL}}$ to the upper and lower levels also enables adaptive balancing of two learning objectives in $\mathcal{L}_{\text{DFCL}} = \mathcal{L}_{\text{DL}} + \alpha\mathcal{L}_{\text{PL}}$, thus eliminating the need for manual hyperparameter tuning of $\alpha$. An overview of the Bi-DFCL framework is shown in Figure 2. Despite these advantages, solving the resulting bi-level optimization problem is non-trivial. The lower-level loss (6) is differentiable with respect to $\theta$, allowing $\mathcal{F}_\theta$ to be updated via gradient descent (GD). However, computing the gradient for the upper-level loss (5) is much more challenging. By the chain rule, we have:

$$\nabla_\phi \mathcal{L}_{\text{DL}}\left(\theta^\star(\phi); \mathcal{D}_{\text{RCT}}\right) = \nabla_\theta \mathcal{L}_{\text{DL}}\left(\theta; \mathcal{D}_{\text{RCT}}\right)|_{\theta=\theta^\star(\phi)} \cdot \frac{\partial\theta^\star(\phi)}{\partial\phi} \tag{10}$$

To calculate the gradient $\nabla_\phi \mathcal{L}_{\text{DL}}(\theta^\star(\phi); \mathcal{D}_{\text{RCT}})$, we require both $\nabla_\theta \mathcal{L}_{\text{DL}}(\theta; \mathcal{D}_{\text{RCT}})$ at $\theta = \theta^\star(\phi)$ and Jacobian $\frac{\partial\theta^\star(\phi)}{\partial\phi}$. However, as will be discussed in Sec.4.2, $\mathcal{L}_{\text{DL}}$ is non-differentiable, and thus

the first term cannot be directly computed. Moreover, the second term is also difficult to obtain, as the optimal solution $\theta^\star(\phi)$ lacks a closed-form expression, making its Jacobian intractable. We will discuss how to address these two non-differentiability challenges in Sec. 4.2 and Sec. 4.4, respectively.

## 4.2 Differentiation of Decision Loss

As is mentioned in Sec. 3, $\mathcal{L}_{\mathrm{DL}}$ is non-computed due to the lack of the counterfactual responses. By leveraging strong ignorability $(X, R(t), C(t)) \perp T$ of experimental data, we derive an unbiased estimator of the decision loss as follows (see Appendix A.2 for the formal proof):

$$\mathcal{L}_{\mathrm{DL}}(\theta; \mathcal{D}_{\mathrm{RCT}}) = -\mathbb{E}_{i,t_i} \left[ \frac{N}{N_{t_i}} \cdot z^*_{it_i}(\hat{r}(\theta), \hat{c}(\theta)) \cdot r_{it_i} \right]. \tag{11}$$

$N_{t_i}$ is the number of individuals assigned treatment $t_i$ in $\mathcal{D}_{\mathrm{RCT}}$, and by the chain rule, the gradient is:

$$\nabla_\theta \mathcal{L}_{\mathrm{DL}}(\theta; \mathcal{D}_{\mathrm{RCT}}) = \frac{\partial \mathcal{L}_{\mathrm{DL}}(\theta; \mathcal{D}_{\mathrm{RCT}})}{\partial z^*_{it_i}(\hat{r}(\theta), \hat{c}(\theta))} \cdot \frac{\partial z^*_{it_i}(\hat{r}(\theta), \hat{c}(\theta))}{\partial \theta}. \tag{12}$$

The first term is trivial since $\mathcal{L}_{\mathrm{DL}}$ is continuously differentiable with respect to $z^*_{it_i}(\hat{r}(\theta), \hat{c}(\theta))$ according to Eq. (11). Based on the Lagrangian relaxation algorithm $\mathcal{A}$ (1), the solution is:

$$z^*_{it_i}(\hat{r}(\theta), \hat{c}(\theta)) = \mathbb{1} \left\{ t_i = \arg\max_{j \in [M]} \left[ \hat{r}_{ij}(\theta) - \lambda^* \hat{c}_{ij}(\theta) \right] \right\} \tag{13}$$

where $\mathbb{1}$ is indicator function and $\lambda^*$ is the optimal Lagrange multiplier. By introducing dual decision variables $z^\lambda_{it_i}(\hat{r}(\theta), \hat{c}(\theta))$ satisfying $z^\lambda_{it_i}(\hat{r}(\theta), \hat{c}(\theta)) = \mathbb{1}_{t_i = \arg\max_{j \in [M]} \hat{r}_{ij}(\theta) - \lambda \hat{c}_{ij}(\theta)}$, the Lagrange multiplier $\lambda^*$ in Eq. (13) can be determined by binary search of $\lambda$ with the terminal condition:

$$\left| \mathbb{E}_{i,t_i} \left[ \frac{N}{N_{t_i}} \cdot z^\lambda_{it_i}(\hat{r}(\theta), \hat{c}(\theta)) \cdot c_{it_i} \right] - \frac{B}{N} \right| \leq \epsilon. \tag{14}$$

Due to the existence of indicator functions, $z^*_{it_i}(\hat{r}(\theta), \hat{c}(\theta))$ is non-differentiable with respect to $\theta$. By utilizing Softmax functions, the discrete solution $z^*_{it_i}(\hat{r}(\theta), \hat{c}(\theta))$ can be relaxed to a continuously differentiable function $z'_{it_i}(\hat{r}(\theta), \hat{c}(\theta))$, which can also be regarded as the probability of $z^*_{it_i} = 1$:

$$z'_{it_i}(\hat{r}(\theta), \hat{c}(\theta)) = \frac{\exp[\hat{r}_{it_i}(\theta) - \lambda^* \hat{c}_{it_i}(\theta)]}{\sum_{j \in [M]} \exp[\hat{r}_{ij}(\theta) - \lambda^* \hat{c}_{ij}(\theta)]}, \tag{15}$$

Hence, we obtain a surrogate decision loss $\mathcal{L}_{\mathrm{PPL}}$ of $\mathcal{L}_{\mathrm{DL}}$, called the primal policy learning loss:

$$\mathcal{L}_{\mathrm{PPL}}(\theta; \mathcal{D}_{\mathrm{RCT}}) = -\mathbb{E}_{i,t_i} \left[ \frac{N}{N_{t_i}} \cdot \frac{\exp[\hat{r}_{it_i}(\theta) - \lambda^* \hat{c}_{it_i}(\theta)]}{\sum_{j \in [M]} \exp[\hat{r}_{ij}(\theta) - \lambda^* \hat{c}_{ij}(\theta)]} \cdot r_{it_i} \right], \tag{16}$$

Note that minimizing $\mathcal{L}_{\mathrm{PPL}}(\theta; \mathcal{D}_{\mathrm{RCT}})$ is equivalent to maximizing the expected reward of policy $\pi = z'_{it_i}(\hat{r}(\theta), \hat{c}(\theta))$. Additionally, an alternative derivation of the primal policy learning loss can be obtained through the maximum entropy regularization trick, as detailed in Appendix A.3. Unlike dual decision loss in [52] which considers all budgets, $\mathcal{L}_{\mathrm{PPL}}$ directly targets decision quality under a specific budget $B$, thereby ensuring better alignment with real-world marketing constraints.

We further introduce the primal improved finite difference strategy (PIFD), which leverages the mathematical definition of the gradient terms $\frac{\partial \mathcal{L}_{\mathrm{DL}}(\theta; \mathcal{D}_{\mathrm{RCT}})}{\partial z'_{ij}(\hat{r}(\theta), \hat{c}(\theta))}$: PIFD directly estimates their values via black-box perturbations on $\mathcal{L}_{\mathrm{DL}}$ and accelerates computation with $\mathcal{L}_{\mathrm{PPL}}$-aware gradient estimator (see Appendix A.4 for details). Compared to $\mathcal{L}_{\mathrm{PPL}}$, PIFD preserves original optimization landscape without relaxation, and by freezing computed gradients as non-trainable nodes, enables seamless integration with automatic differentiation libraries. This final surrogate decision loss $\mathcal{L}_{\mathrm{PIFD}}$ is given:

$$\mathcal{L}_{\mathrm{PIFD}}(\theta; \mathcal{D}_{\mathrm{RCT}}) = \mathbb{E}_{i \in [N], j \in [M]} \left[ \frac{\partial \mathcal{L}_{\mathrm{DL}}(\theta; \mathcal{D}_{\mathrm{RCT}})}{\partial z'_{ij}(\hat{r}(\theta), \hat{c}(\theta))} \cdot z'_{ij}(\hat{r}(\theta), \hat{c}(\theta)) \right]. \tag{17}$$

### 4.3 Implicit Differentiation-Based Algorithm

Next, we address the second challenge in Bi-DFCL: computing the Jacobian $\frac{\partial \theta^\star(\phi)}{\partial \phi}$ without a closed-form solution for $\theta^\star(\phi)$, a well-known issue in BLO. A common approach is to explicitly differentiate through the gradient descent step, assuming $\theta^\star(\phi)$ can be reached in one GD step [15, 9, 41] (see Appendix A.5). However, this method relies on the optimization path and, when combined with decision loss, often suffers from vanishing gradients and suboptimal solutions. To address this, we propose an implicit differentiation-based algorithm. Note that the optimal solution $\theta^\star(\phi)$ satisfies the first-order condition: $\frac{\partial \mathcal{L}_{\mathrm{PL}}(\phi, \theta; \mathcal{D}_{\mathrm{OBS}})}{\partial \theta}|_{\theta = \theta^\star(\phi)} = 0$. Differentiating both sides with respect to $\phi$ gives:

$$\frac{\partial^2 \mathcal{L}_{\mathrm{PL}}(\phi, \theta; \mathcal{D}_{\mathrm{OBS}})}{\partial \theta^2}|_{\theta = \theta^\star(\phi)} \cdot \frac{\partial \theta^\star(\phi)}{\partial \phi} = -\frac{\partial^2 \mathcal{L}_{\mathrm{PL}}(\phi, \theta; \mathcal{D}_{\mathrm{OBS}})}{\partial \phi \partial \theta}|_{\theta = \theta^\star(\phi)} \quad (18)$$

Eq. (18) is also a direct result of the implicit function theorem [34]. Notably, this approach avoids explicitly storing the optimization trajectory; the optimal solution $\theta^\star(\phi)$ can be obtained using any optimization algorithm, and we only need to differentiate the optimality condition it satisfies to implicitly obtain its Jacobian. This path-independence leads to more accurate and stable gradients.

While a closed-form expression for the Jacobian $\frac{\partial \theta^\star(\phi)}{\partial \phi}$ can be directly derived, computing and storing the inverse of the Hessian matrix is computationally expensive, especially in large-scale marketing applications. To overcome this, we employ the conjugate gradient (CG) algorithm [34], which solves $Ax = b$ by equivalently minimizing $\frac{1}{2}x^\top A x - b^\top x$ and can be implemented using only Hessian-vector products. This approach efficiently solves (18) without explicit Hessian construction or inversion (see Appendix A.6), making Bi-DFCL applicable to large-scale marketing optimization.

### 4.4 Overall training procedure of Bi-DFCL

We now summarize the overall training procedure of Bi-DFCL in Algorithm 1.

---

**Algorithm 1** Pseudocode for Bi-Level Decision-Focused Causal Learning (Bi-DFCL)

---

**Input:** $\mathcal{D}_{\mathrm{RCT}} \leftarrow \{(x_i, t_i, r_{it_i}, c_{it_i})\}_{i=1}^{N_{\mathrm{RCT}}}$, $\mathcal{D}_{\mathrm{OBS}} \leftarrow \{(x_i, t_i, r_{it_i}, c_{it_i})\}_{i=1}^{N_{\mathrm{OBS}}}$, Target Network $\mathcal{F}_\theta$, Bridge Network $\mathcal{G}_\phi$, Teacher Network $\mathcal{F}_\psi$, $k$ (number of GD steps for assumed updates, default $k = 5$).
**Pretrain** Teacher Network $\mathcal{F}_\psi$ on $\mathcal{D}_{\mathrm{RCT}}$ using any uplift model with standard MSE loss
**Initialize** Target Network $\mathcal{F}_\theta$ (random or warm start) and Bridge Network $\mathcal{G}_\phi$ (random).

1: **for** each mini-batch $\mathcal{B}_{\mathrm{OBS}}^{(b)}$ in $\mathcal{D}_{\mathrm{OBS}}$ over all epochs **do**
2:      **if** $b \bmod k = 0$ **(i.e., every $k$-th batch), then solve the upper-level problem** (5) **:**
3:          **Step 1** —— Perform $k$ assumed updates to obtain $\theta^\star(\phi)$ (without modifying $\mathcal{F}_\theta$):
4:              Copy $\mathcal{F}_\theta$ to $\mathcal{F}'_\theta$; generate counterfactual pseudo-labels $r_{i,j}^{\mathrm{cf}}, c_{i,j}^{\mathrm{cf}}$ for $\mathcal{B}_{\mathrm{OBS}}^{(b)}$ as in Eq. (7)–(8).
5:              Perform $k$ steps gradient descent(GD) on $\mathcal{L}_{\mathrm{PL}}(\phi, \theta; \mathcal{B}_{\mathrm{OBS}}^{(b)})$ (Eq. (9)) so that $\mathcal{F}'_\theta$ update to $\mathcal{F}'_{\theta_\star}$.
6:          **Step 2** —— Obtain two non-differentiability terms as shown in Eq. (10):
7:              Solve Eq. (18) via conjugate gradient (CG) Algorithm to obtain Jacobian $\frac{\partial \theta^\star(\phi)}{\partial \phi}$.
8:              Using $\mathcal{F}'_{\theta_\star}$, compute $\mathcal{L}_{\mathrm{PPL}}16$ or $\mathcal{L}_{\mathrm{PIFD}}17$ on $\mathcal{D}_{\mathrm{RCT}}$ , obtain $\nabla_\theta \mathcal{L}_{\mathrm{DL}}(\theta; \mathcal{D}_{\mathrm{RCT}})|_{\theta = \theta^\star(\phi)}$.
9:          **Step 3** —— End-to-End update Bridge Network $\mathcal{G}_\phi$ according to Eq.(10):
10:              Perform one GD step on $\mathcal{G}_\phi$ with $\mathcal{D}_{\mathrm{RCT}}$ : $\phi \leftarrow \phi - \alpha_\phi \cdot \nabla_\theta \mathcal{L}_{\mathrm{DL}}(\theta; \mathcal{D}_{\mathrm{RCT}})|_{\theta = \theta^\star(\phi)} \cdot \frac{\partial \theta^\star(\phi)}{\partial \phi}$.
11:      **end if**
12:      **Solve the lower-level problem** (6) **with the latest $\mathcal{G}_\phi$:**
13:          Generate updated counterfactual pseudo-labels $r_{i,j}^{\mathrm{cf}}, c_{i,j}^{\mathrm{cf}}$ for $\mathcal{B}_{\mathrm{OBS}}^{(b)}$ as in Eq. (7)–(8).
14:          Compute $\mathcal{L}_{\mathrm{PL}}(\phi, \theta; \mathcal{B}_{\mathrm{OBS}}^{(b)})$ (Eq. (9)); update $\mathcal{F}_\theta$ by one GD step: $\theta \leftarrow \theta - \alpha_\theta \cdot \nabla_\theta \mathcal{L}_{\mathrm{PL}}(\phi, \theta; \mathcal{D}_{\mathrm{OBS}})$.
15: **end for**
     **Output:** Well-trained Target Network $\mathcal{F}_\theta$ for predicting $\hat{r}_{ij}, \hat{c}_{ij}$.

---

## 5 Real-World Experiments

### 5.1 Offline Experimental Setup

**Dataset and Preprocessing.** Three types of offline datasets are provided: an open real-world dataset and two marketing datasets collected from Meituan, an online food delivery platform. The detailed statistics of three datasets are shown in Table 1. Readers can see more details in Appendix B.1.

- **CRITEO-UPLIFT v2.** This public dataset from Criteo [11] contains 13.9 million RCT samples, each with 12 features, a binary treatment indicator, and two response labels (visit/conversion). Since practical marketing scenarios typically have large number of OBS data and little RCT data, we simulate a marketing policy to convert part of the RCT data into OBS data. Further details can be found in Appendix B.1.1. We refer to the transformed dataset as CRITEO-UPLIFT v2 (Hybrid).

- **Marketing data I.** Money-off is a common marketing campaign at Meituan, an online food delivery platform. We conduct a two-month RCT to collect data in this platform. The money-off $T \in \{0, 1, \ldots, 7\}$ is taken as the treatment, where $T = t$ means $\$t$ cash off for each order whose price meets a given threshold. This dataset contains 180 features, 1 treatment label and 2 response labels (daily cost/orders). This dataset contains 5.5 million RCT and 22.2 million OBS samples.

- **Marketing data II.** Discounting is another common marketing campaign at Meituan. We conduct a four-week RCT to collect data. The discount $T \in \{0, 5, 10, 15, 20\}$ is taken as the treatment, where $T = t$ means $t\%$ off for each order whose price meets a given threshold. This dataset contains 192 features, 1 treatment label and 2 response labels (daily cost/orders). This dataset contains 5.0 million RCT samples and 33.8 million OBS samples.

Table 1: Statistics of three offline datasets.

| Dataset | Features | Treatment | Training (OBS) | Training (RCT) | Validation (RCT) | Test (RCT) |
|---|---|---|---|---|---|---|
| CRITEO-UPLIFT v2 (Hybrid) | 12 | 2 | 3498294 | 698980 | 1397959 | 4193878 |
| Marketing data I | 180 | 8 | 22201405 | 2220781 | 555014 | 2775976 |
| Marketing data II | 192 | 5 | 33815274 | 2017450 | 504362 | 2521813 |

**Baselines and Experimental Details.** We compare the proposed methods with three categories of causal learning baselines: (1) Methods trained with RCT data, (2) Methods trained with OBS data, and (3) Methods trained with both RCT and OBS data. Also see more details in Appendix B.1.3.

- **Methods trained with RCT data**: With RCT data only, the baselines include two simple two-stage methods: TSM-SL[48], TSM-CF[2], and three end-to-end methods: DHCL[51], DFCL-DPL[52], DFCL-DIFD[52]. Note that these end-to-end methods can only be trained using RCT data.

- **Methods trained with OBS data**: With OBS data only, the baselines include two simple two-stage methods: TSM-SL[48], TSM-CF[2], and two reweighting-based methods: IPS[30], DR-JT[40], and three representation learning methods: CFR-WASS[33], CFR-MMD[33], DragonNet[35].

- **Methods trained with both RCT and OBS data**: Based on both RCT and OBS data, the baselines include TSM-SL[48], and reweighting-based methods: LTD-IPS[41], LTD-DR[41], AutoDebias[9], and representation learning methods: CausE[8], KD-Label[19], KD-Feature[19].

**Evaluation Metrics.** Two evaluation metrics are provided for offline evaluation in this experiment.

- **AUCC (Area under Cost Curve).** A common metric used in existing works [2, 13, 51], which is designed for evaluating the performance to rank ROI of individuals in the binary treatment setting. Because AUCC represents the decision quality of marketing under binary treatments, we use AUCC to compare the performance of different methods in CRITEO-UPLIFT v2(Hybrid).

- **EOM (Expected Outcome Metric).** EOM is also commonly used in [2, 51, 49, 52]. Based on RCT data, an unbiased estimation of the expected outcome (per-capita revenue/per-capita cost) for arbitrary budget allocation policy can be obtained. Details of EOM are shown in Appendix B.1.2. Since EOM represents the decision quality of marketing under multilple treatments, we use EOM to compare the performance of different methods in Marketing data I and II.

## 5.2 Offline Experimental Results

**Overall Performance Comparison.** Table 2 compares Bi-DFCL with all baselines. We have four main observations: **(1)**: Among methods trained solely on RCT data, end-to-end methods consistently outperform two-stage methods across all datasets, highlighting the importance of directly optimizing for decision quality and validating our motivation to bridge the prediction-decision gap. **(2)**: Our proposed DFCL-PPL and DFCL-PIFD outperform dual decision loss, showing that optimizing primal decision losses better aligns with real-world marketing constraints, as they directly target decision quality under specific budget values $B$. **(3)**: The relative performance of TSM trained on RCT or

Table 2: Performances of the proposed methods and baselines (mean $\pm$ standard deviation across 20 runs). The best result is bolded and the best results of three types of baseline methods are underlined.

| Data | Methods | CRITEO-UPLIFT v2 (Hybrid) | | Marketing Data I | | Marketing Data II | |
|------|---------|------|-------------|-----|-------------|-----|-------------|
| | | AUCC | Improvement | EOM | Improvement | EOM | Improvement |
| RCT | TSM-SL | 0.7143±0.0299 | – | 1.0000±0.0032 | – | 1.0000±0.0020 | – |
| RCT | TSM-CF | 0.6730±0.0196 | -5.78% | 0.9767±0.0005 | -2.33% | 0.9680±0.0006 | -3.20% |
| RCT | DHCL | 0.7278±0.0358 | 1.90% | 0.9972±0.0011 | -0.28% | 1.0059±0.0007 | 0.59% |
| RCT | DFCL-DPL | 0.7416±0.0170 | 3.82% | 1.0120±0.0020 | 1.20% | 1.0094±0.0008 | 0.94% |
| RCT | DFCL-DIFD | 0.7441±0.0233 | 4.17% | 1.0151±0.0033 | 1.51% | 1.0110±0.0029 | 1.10% |
| RCT | **DFCL-PPL (Ours)** | 0.7419±0.0128 | 3.86% | 1.0167±0.0024 | 1.67% | 1.0156±0.0016 | 1.56% |
| RCT | **DFCL-PIFD (Ours)** | 0.7437±0.0204 | 4.12% | 1.0170±0.0024 | 1.70% | 1.0153±0.0016 | 1.53% |
| OBS | TSM-SL | 0.7413±0.0038 | 3.78% | 1.0067±0.0013 | 0.67% | 0.9957±0.0015 | -0.43% |
| OBS | TSM-CF | 0.7105±0.0020 | -0.53% | 0.9825±0.0002 | -1.75% | 0.9680±0.0004 | -3.20% |
| OBS | IPS | 0.7092±0.0131 | -0.71% | 1.0070±0.0037 | 0.70% | 0.9990±0.0026 | -0.10% |
| OBS | DR-JT | 0.7439±0.0053 | 4.14% | 1.0102±0.0019 | 1.02% | 1.0054±0.0018 | 0.54% |
| OBS | CFR-WASS | 0.7245±0.0109 | 1.43% | 1.0032±0.0020 | 0.32% | 0.9961±0.0013 | -0.39% |
| OBS | CFR-MMD | 0.7339±0.0045 | 2.74% | 1.0055±0.0020 | 0.55% | 0.9997±0.0033 | -0.03% |
| OBS | DragonNet | 0.7490±0.0066 | 4.86% | 1.0069±0.0041 | 0.69% | 0.9988±0.0021 | -0.12% |
| RCT+OBS | TSM-SL | 0.7438±0.0032 | 4.13% | 1.0071±0.0011 | 0.71% | 0.9988±0.0022 | -0.12% |
| RCT+OBS | CausE | 0.7392±0.0081 | 3.49% | 1.0031±0.0019 | 0.31% | 1.0001±0.0014 | 0.01% |
| RCT+OBS | KD-Label | 0.7374±0.0055 | 3.23% | 1.0033±0.0027 | 0.33% | 0.9997±0.0019 | -0.03% |
| RCT+OBS | KD-Feature | 0.7306±0.0064 | 2.28% | 1.0074±0.0019 | 0.74% | 0.9983±0.0019 | -0.17% |
| RCT+OBS | LTD-IPS | 0.7427± 0.0080 | 3.98% | 1.0120±0.0036 | 1.20% | 1.0040±0.0042 | 0.40% |
| RCT+OBS | LTD-DR | 0.7533± 0.0059 | 5.46% | 1.0168±0.0026 | 1.68% | 1.0067±0.0021 | 0.67% |
| RCT+OBS | AutoDebias | 0.7489± 0.0077 | 4.84% | 1.0175±0.0027 | 1.75% | 1.0066±0.0032 | 0.66% |
| RCT+OBS | **Bi-DFCL-PPL (Ours)** | 0.7797± 0.0094 | 9.16% | 1.0277±0.0024 | 2.77% | **1.0252±0.0023** | **2.52%** |
| RCT+OBS | **Bi-DFCL-PIFD (Ours)** | **0.7812± 0.0084** | **9.37%** | **1.0297±0.0030** | **2.97%** | 1.0249±0.0018 | 2.49% |

OBS data varies across datasets, reflecting the complementary strengths of the two data sources: RCT data offer low bias but high variance, while OBS data are more biased but lower variance. However, all existing end-to-end methods are restricted to RCT data, limiting their ability to leverage abundant OBS data and making them prone to overfitting. **(4)**: Bi-DFCL consistently outperforms all baselines on all datasets, demonstrating superior alignment of prediction and decision objectives and ability to achieve optimal bias-variance tradeoff by fully leveraging both RCT and OBS data. By overcoming the overreliance on limited RCT data that hampers previous decision-focused methods, Bi-DFCL delivers improved generalization and decision quality in real-world marketing scenarios.

**Ablation Studies.** To show the effects of individual components, we conduct ablation study by incrementally adding four key components of Bi-DFCL to baseline in a sequential manner: Decision Loss (PPL), Bi-level Optimization by hybrid RCT and OBS data, Counterfactual Labels, and Implicit Differentiation Algorithm. The experimental results on marketing datasets are reported in Table 3. We can find that after the introduction of each module, the performance can all be strengthened to some extent, which demonstrates that our three contributions can all benefit the marketing optimization. In addition, we provide detailed descriptions for these baselines of ablation studies in Appendix B.2.

Table 3: Ablation study of each individual component in Bi-DFCL with two marketing datasets.

| Components of Bi-DFCL | | | | Marketing Data I | | Marketing Data II | |
|-----------------------|----------------------|----------------------|--------------------------|------|-------------|------|-------------|
| Decision Loss (PPL) | Bi-level Optimization | Counterfactual Labels | Implicit Differentiation | EOM | Improvement | EOM | Improvement |
| ✗ | ✗ | ✗ | ✗ | 1.0000 | – | 1.0000 | – |
| ✓ | ✗ | ✗ | ✗ | 1.0167 | 1.67% | 1.0156 | 1.56% |
| ✓ | ✓ | ✗ | ✗ | 1.0240 | 2.40% | 1.0175 | 1.75% |
| ✓ | ✓ | ✓ | ✗ | 1.0248 | 2.48% | 1.0213 | 2.13% |
| ✓ | ✓ | ✓ | ✓ | **1.0277** | **2.77%** | **1.0252** | **2.52%** |

**In-depth Analysis.** We conduct in-depth analysis to explore the effect of the training data size, as well as to validate the bias-variance properties of the RCT and OBS data. We further evaluate the sensitivity of the hyper-parameters using different values and evaluate the robustness of our proposed methods under multiple sets of budget values $B$. Additionally, we also provide a detailed discussion comparing the computational overhead of Bi-DFCL against different baseline methods. See Appendix B.3 for more detailed experimental results.

## 5.3 Online A/B Tests

**Setups.** We deploy our proposed Bi-DFCL-PPL, Bi-DFCL-PIFD and three baselines: DFCL-PIFD, LTD-DR and TSM-SL together to support a discount campaign at Meituan (our online food delivery platform) and conduct large-scale online A/B tests for four weeks. The experiment contains 790K

online shops and they are randomly divided every day into five groups called G-BPPL, G-BPIFD, G-PIFD , G-LTD and G-TSL respectively. Each shop will be assigned a discount $t \in \{0, 5, 10, 15, 20\}$ as the treatmemt, which means $t\%$ off for each order whose price meets a given threshold. The marketing goal is to maximize the orders by assigning an appropriate discount to each store every day for a limited budget that may change slightly from day to day.

Table 4: Results of online A/B tests with the confidence interval (four weeks)

| Method | Group | Week | | | | Improvement |
|---|---|---|---|---|---|---|
| | | 1st | 2nd | 3rd | 4th | |
| TSM-SL | G-TSL | $1.0000 \pm 0.0022$ | $1.0335 \pm 0.0030$ | $0.9217 \pm 0.0017$ | $0.9720 \pm 0.0048$ | – |
| LTD-DR | G-LTD | $1.0183 \pm 0.0020$ | $1.0378 \pm 0.0039$ | $0.9344 \pm 0.0037$ | $0.9723 \pm 0.0070$ | 0.91% |
| DFCL-PIFD | G-PIFD | $1.0302 \pm 0.0013$ | $1.0436 \pm 0.0020$ | $0.9440 \pm 0.0020$ | $0.9799 \pm 0.0018$ | 1.80% |
| Bi-DFCL-PPL | G-BPPL | $1.0428 \pm 0.0019$ | $1.0558 \pm 0.0025$ | $0.9582 \pm 0.0019$ | $0.9872 \pm 0.0014$ | 3.00% |
| Bi-DFCL-PIFD | G-BPIFD | $1.0470 \pm 0.0021$ | $1.0537 \pm 0.0027$ | $0.9581 \pm 0.0024$ | $0.9906 \pm 0.0031$ | 3.22% |

**Results.** Table 4 illustrates the online weekly orders for five groups during four weeks. In order to preserve data privacy, all data points have been normalized that are divided by the orders of TSM-SL in the first week. We can see that Bi-DFCL exhibits significantly superior overall performance during four weeks, which validates the effectiveness of Bi-DFCL for real-world marketing optimization.

## 6 Conclusion

In this paper, we propose the Bi-Level Decision-Focused Causal Learning (Bi-DFCL) framework for large-scale marketing optimization, addressing two key challenges in existing approaches: prediction-decision misalignment and bias-variance dilemma. Extensive offline experiments and online A/B tests demonstrate that Bi-DFCL consistently outperforms state-of-the-art. Our future work includes further improving computational efficiency and applying Bi-DFCL to other decision-making domains.

## Acknowledgments

This work was supported in part by the NSF of China (62422207).

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

# A More details of Bi-DFCL Framework

Here, we provide additional details for Sec.3 and Sec.4 of the main text.

## A.1 The Lagrangian relaxation algorithm $\mathcal{A}$ in Sec.3

We give pseudocode of the Lagrangian relaxation algorithm $\mathcal{A}$ in Algorithm 2.

---

**Algorithm 2** The Lagrangian relaxation algorithm $\mathcal{A}$ for the primal formulation of MTBAP

---

**Input:** budget $B$; predicted revenue/cost $\hat{r}, \hat{c}$; data $D \equiv \{(x_i, t_i, r_{it_i}, c_{it_i})\}_{i=1}^N$; small constant $\epsilon$.
**Compute:** For each $t$ in $[M]$, $N_t \leftarrow$ number of samples with $t_i = t$; $p_t \leftarrow N_t/N$.

**Initialize:** $\lambda_{\min} \leftarrow 0$, $\lambda_{\max} \leftarrow \max_{i,j}\left(\frac{\hat{r}_{ij}}{\hat{c}_{ij}}\right)$, $z_{ij} \leftarrow 0$ for all $i, j$

1: **while** $\lambda_{\max} - \lambda_{\min} > \epsilon$ **do**
2:      $\lambda \leftarrow \frac{\lambda_{\max} + \lambda_{\min}}{2}$
3:      **for all** $i, j$ **do**
4:          $z_{ij} \leftarrow \mathbb{I}\left(j = \arg\max_j(\hat{r}_{ij} - \lambda\hat{c}_{ij})\right)$
5:      **end for**
6:      $\bar{c}(\lambda, r, c, \hat{r}, \hat{c}) \leftarrow \frac{1}{N}\sum_i \frac{1}{p_{t_i}}c_{t_i}\mathbb{I}(t_i = \arg\max_j z_{ij})$
7:      **if** $\left|\frac{B}{N} - \bar{c}(\lambda, r, c, \hat{r}, \hat{c})\right| < \epsilon$ **then**
8:          **break**
9:      **end if**
10:     **if** $\frac{B}{N} - \bar{c}(\lambda, r, c, \hat{r}, \hat{c}) > 0$ **then**
11:        $\lambda_{\max} \leftarrow \lambda$
12:     **else**
13:        $\lambda_{\min} \leftarrow \lambda$
14:     **end if**
15: **end while**
16: $\lambda^* \leftarrow \lambda$
     **Output:** Solution $z_{ij}$ for MTBAP with a worst-case approximation ratio of $\rho = 1 - \frac{\max_{ij} r_{ij}}{\text{OPT}}$

---

## A.2 The Formal Proof of Eq. (11) in Sec.4.2

*Proof.* Recall that Eq. 11 is given by $\mathcal{L}_{\text{DL}}(\theta; \mathcal{D}_{\text{RCT}}) = -\mathbb{E}_{i,t_i}\left[\frac{N}{N_{t_i}} \cdot z_{it_i}^*(\hat{r}(\theta), \hat{c}(\theta)) \cdot r_{it_i}\right]$. We aim to show that this is an unbiased estimator of $\mathcal{L}_{\text{DL}}(\theta) = -M \cdot \mathbb{E}_{i\in[N], j\in[M]}\left[z_{ij}^*(\hat{r}(\theta), \hat{c}(\theta)) \cdot r_{ij}\right]$. Note that $\mathcal{L}_{\text{DL}}(\theta; \mathcal{D}_{\text{RCT}})$ can be rewritten as:

$$
\begin{aligned}
\mathcal{L}_{\text{DL}}(\theta; \mathcal{D}_{\text{RCT}}) &= -\mathbb{E}_{i,t_i}\left[\frac{N}{N_{t_i}} \cdot z_{it_i}^*(\hat{r}(\theta), \hat{c}(\theta)) \cdot r_{it_i}\right] \\
&= -\frac{1}{N}\sum_i \frac{N}{N_{t_i}} \cdot z_{it_i}^*(\hat{r}(\theta), \hat{c}(\theta)) \cdot r_{it_i} \\
&= -\sum_j \sum_{i:t_i=j} \frac{1}{N_{t_i}} \cdot z_{it_i}^*(\hat{r}(\theta), \hat{c}(\theta)) \cdot r_{it_i} \\
&= -\sum_j \mathbb{E}_i[z_{it_i}^*(\hat{r}(\theta), \hat{c}(\theta)) \cdot r_{it_i} | t_i = j] \\
&= -\sum_j \mathbb{E}_i[z_{ij}^*(\hat{r}(\theta), \hat{c}(\theta)) \cdot r_{ij}] \quad (T \perp X) \\
&= M \cdot -\mathbb{E}_{i\in[N], j\in[M]}\left[z_{ij}^*(\hat{r}(\theta), \hat{c}(\theta)) \cdot r_{ij}\right].
\end{aligned}
$$

where $T \perp X$ holds because the data set is $\mathcal{D}_{\text{RCT}}$ from random control trials (RCT). Therefore,

$$
-\mathbb{E}_{i,t_i}\left[\frac{N}{N_{t_i}}z_{it_i}^*(\hat{r}(\theta), \hat{c}(\theta)) \cdot r_{it_i}\right] = M \cdot -\mathbb{E}_{i\in[N], j\in[M]}\left[z_{ij}^*(\hat{r}(\theta), \hat{c}(\theta)) \cdot r_{ij}\right],
$$

which completes the proof. Note that there is a multiplicative factor of $M$. To ensure consistency, we revise Eq. 3 in the main text as : $\mathcal{L}_{\text{DL}}(\theta) = -M \cdot \mathbb{E}_{i\in[N], j\in[M]}\left[z_{ij}^*(\hat{r}(\theta), \hat{c}(\theta)) \cdot r_{ij}\right]$. $\square$

## A.3 More details of The maximum entropy regularization trick in Sec.4.2

The primal policy learning loss $\mathcal{L}_{\text{PPL}}$ can also be derived through the maximum entropy regularization trick. After obtaining the optimal Lagrange multiplier $\lambda^*$ via binary search, we introduce a maximum entropy regularizer into the objective function of the dual formulation of the MTBAP:

$$\max_z \sum_i \sum_j (\hat{r}_{ij} - \lambda^* \hat{c}_{ij}) z_{ij} - \tau \sum_i \sum_j z_{ij} \ln z_{ij},$$

$$s.t. \quad \sum_j z_{ij} = 1, \forall i,$$

$$z_{ij} \in [0,1],$$

where $\tau > 0$ is the temperature hyperparameter controlling the entropy regularization strength. Note that the dual formulation of MTBAP can be equivalently relaxed to $z \in [0,1]$. Further we have:

$$L(z,\beta) = \sum_{i=1}^{N} \sum_{j=1}^{M} (\hat{r}_{ij} - \lambda^* \hat{c}_{ij}) z_{ij} - \tau \sum_{i=1}^{N} \sum_{j=1}^{M} z_{ij} \ln z_{ij} - \sum_i \beta_i \left(1 - \sum_j z_{ij}\right), \quad (19)$$

where $\beta$ represents the dual variables associated with the equality constraints. Setting $\frac{\partial L(z,\beta)}{\partial z} = 0$ and $\frac{\partial L(z,\beta)}{\partial \beta} = 0$ yields the optimal solution:

$$z_{ij}^d = \frac{\exp\left[(\hat{r}_{ij} - \lambda^* \hat{c}_{ij})/\tau\right]}{\sum_k \exp[(\hat{r}_{ik} - \lambda^* \hat{c}_{ik})/\tau]}. \quad (20)$$

Substituting this into Eq.11 , we derive $\mathcal{L}_{\text{PPL}}$ by the maximum entropy regularization trick:

$$\mathcal{L}_{\text{PPL}}(\theta; \mathcal{D}_{\text{RCT}}) = -\mathbb{E}_{i,t_i} \left[ \frac{N}{N_{t_i}} \cdot \frac{\exp[(\hat{r}_{it_i}(\theta) - \lambda^* \hat{c}_{it_i}(\theta))/\tau]}{\sum_{j \in [M]} \exp[(\hat{r}_{ij}(\theta) - \lambda^* \hat{c}_{ij}(\theta))/\tau]} \cdot r_{it_i} \right] \quad (21)$$

## A.4 More details of The Primal Improved Finite Difference Strategy (PIFD) in Sec.4.2

The primal improved finite difference strategy (PIFD) estimates gradients $\frac{\partial \mathcal{L}_{\text{DL}}(\theta; \mathcal{D}_{\text{RCT}})}{\partial z'_{ij}(\hat{r}(\theta), \hat{c}(\theta))}$ via black-box perturbations on $\mathcal{L}_{\text{DL}}$. Using the finite difference strategy, the gradient of $\mathcal{L}_{\text{DL}}(\theta; \mathcal{D}_{\text{RCT}})$ with respect to $\hat{r}_{ij}$ is estimated as:

$$\frac{\partial \mathcal{L}_{DL}(r,c,\hat{r},\hat{c})}{\partial \hat{r}_{ij}} = \frac{\mathcal{L}_{DL}(r,c,\hat{r}+e_{ij}h,\hat{c}) - \mathcal{L}_{DL}(r,c,\hat{r},\hat{c})}{h},$$

where $h$ is a small constant, and $e_{ij} \in \{0,1\}^{N \times M}$ is a matrix where only the element in the i-th row and j-th column is 1, and all other elements are 0. The gradient term $\frac{\partial \mathcal{L}_{DL}(r,c,\hat{r},\hat{c})}{\partial \hat{c}_{ij}}$ can be computed similarly. We accelerate above computation with the $\mathcal{L}_{\text{PPL}}$-aware gradient estimator. This involves two key improvements: first, replacing the black-box perturbation with a semi-black-box one; and second, unifying the separate perturbations on $r$ and $c$ into a single perturbation on $z$. Together, these changes improve the stability of the gradient and significantly accelerate the solution process. Given the gradients $\frac{\partial \mathcal{L}_{\text{DL}}(\theta; \mathcal{D}_{\text{RCT}})}{\partial z'_{ij}(\hat{r}(\theta), \hat{c}(\theta))}$ and freeze them, this final surrogate decision loss $\mathcal{L}_{\text{PIFD}}$ is defined as:

$$\mathcal{L}_{\text{PIFD}}(\theta; \mathcal{D}_{\text{RCT}}) = \mathbb{E}_{i \in [N], j \in [M]} \left[ \frac{\partial \mathcal{L}_{\text{DL}}(\theta; \mathcal{D}_{\text{RCT}})}{\partial z'_{ij}(\hat{r}(\theta), \hat{c}(\theta))} \cdot z'_{ij}(\hat{r}(\theta), \hat{c}(\theta)) \right]$$

$$= \mathbb{E}_{i \in [N], j \in [M]} \left[ \frac{\partial \mathcal{L}_{\text{DL}}(\theta; \mathcal{D}_{\text{RCT}})}{\partial z'_{ij}(\hat{r}(\theta), \hat{c}(\theta))} \cdot \frac{\exp[\hat{r}_{ij}(\theta) - \lambda^* \hat{c}_{ij}(\theta)]}{\sum_{j' \in [M]} \exp[\hat{r}_{ij'}(\theta) - \lambda^* \hat{c}_{ij'}(\theta)]} \right]$$

The pseudocode for the $\mathcal{L}_{\text{PPL}}$-aware gradient estimator in PIFD is provided in Algorithm 3. For each sample, we first compute the minimal perturbation that alters the primal decision loss, and then update the loss by correcting only the original result. For clarity, Algorithm 3 is presented using for loops; in practice, we implement it with matrix operations in order to accelerates computation.

**Algorithm 3** $\mathcal{L}_{\text{PPL}}$-aware gradient estimator of the primal improved finite difference strategy (PIFD)

---

**Input:** budget $B$; Training data $D \equiv \{(x_i, t_i, r_{it_i}, c_{it_i})\}_{i=1}^N$; predicted revenue/cost $\hat{r}, \hat{c}$.
**Compute:** For each $t$ in $[M]$, $N_t \leftarrow$ number of samples with $t_i = t$; $p_t \leftarrow N_t/N$.
**Initialize:** $\frac{\partial \mathcal{L}_{\text{DL}}(\theta; \mathcal{D}_{\text{RCT}})}{\partial z'_{ij}(\hat{r}(\theta), \hat{c}(\theta))} = 0$, $z_{ij} = 0$ for all $i, j$.
**Given** $\hat{r}, \hat{c}$, and $D$, **call Algorithm 2** to obtain $\lambda^*$ and $z_{ij}$.

1: $\forall i, j$, $a_{ij} = (r_{ij} - \lambda^* \cdot c_{ij})$, $z_{ij} = \mathbb{I}_{j = \arg\max_j (r_{ij} - \lambda^* \cdot c_{ij})}$
2: $\bar{r}(\lambda^*, r, c, \hat{r}, \hat{c}) \leftarrow \frac{1}{N} \sum_i \frac{1}{p_{t_i}} r_{t_i} \mathbb{I}_{t_i = \arg\max_j z_{ij}}$, $-\mathcal{L}_{DL}(B, r, c, \hat{r}, \hat{c}) \leftarrow \bar{r}(\lambda^*, r, c, \hat{r}, \hat{c})$
3: matching_indices $= \{i \mid t_i = \arg\max_j z_{ij}, \ \forall i\}$
4: mismatching_indices $= \{i \mid t_i \neq \arg\max_j z_{ij}, \ \forall i\}$
5: **for all** $i \in$ matching_indices **do**
6: $\quad h^z_{it_i} = \max_{j \neq t_i} a_{ij} - a_{it_i}$
7: $\quad \frac{\partial - \mathcal{L}_{DL}}{\partial z'_{it_i}} = \frac{-\frac{1}{N \cdot p_{t_i}} \cdot r_{it_i}}{h^z_{it_i}}$
8: $\quad$ **for all** $j \in \{1, 2, ..., M\}$, $j \neq t_i$ **do**
9: $\qquad h^z_{ij} = a_{it_i} - a_{ij}$
10: $\qquad \frac{\partial - \mathcal{L}_{DL}}{\partial z'_{ij}} = \frac{-\frac{1}{N \cdot p_{t_i}} \cdot r_{it_i}}{h^z_{ij}}$
11: $\quad$ **end for**
12: **end for**
13: **for all** $i \in$ mismatching_indices **do**
14: $\quad j = \arg\max_j a_{ij}$
15: $\quad h^z_{it_i} = a_{ij} - a_{it_i}$, $h^z_{ij} = -h^r_{it_i}$
16: $\quad \frac{\partial - \mathcal{L}_{DL}}{\partial z'_{it_i}} = \frac{\frac{1}{N \cdot p_{t_i}} \cdot r_{it_i}}{h^z_{it_i}}$
17: $\quad \frac{\partial - \mathcal{L}_{DL}}{\partial z'_{ij}} = \frac{\frac{1}{N \cdot p_{t_i}} \cdot r_{it_i}}{h^z_{ij}}$
18: **end for**
**Output:** the gradients $\frac{\partial \mathcal{L}_{\text{DL}}(\theta; \mathcal{D}_{\text{RCT}})}{\partial z'_{ij}(\hat{r}(\theta), \hat{c}(\theta))} = -\frac{\partial - \mathcal{L}_{\text{DL}}(\theta; \mathcal{D}_{\text{RCT}})}{\partial z'_{ij}(\hat{r}(\theta), \hat{c}(\theta))}$; the optimal Lagrange multiplier $\lambda^*$.

---

## A.5 The Explicit Differentiation Algorithm for Bi-level Optimization in Sec.4.3

We now introduce the explicit differentiation algorithm for Bi-level Optimization. As discussed in Sec. 4.3, computing the Jacobian $\frac{\partial \theta^\star(\phi)}{\partial \phi}$ in the absence of a closed-form solution for $\theta^\star(\phi)$ is a well-known challenge in bilevel optimization (BLO). A common approach is to explicitly differentiate through the gradient descent step, under the assumption that $\theta^\star(\phi)$ can be reached in a single gradient descent (GD) step [15, 9, 41], as shown below:

$$\theta^*(\phi) \leftarrow \theta - \alpha_\theta \cdot \nabla_\theta \mathcal{L}_{\text{PL}}(\phi, \theta; \mathcal{D}_{\text{OBS}}). \tag{22}$$

By retaining the above update path within any automatic differentiation library, we can explicitly differentiate through the gradient step to compute gradients with respect to the bridge model parameters $\phi$. Specifically, $\nabla_\phi \mathcal{L}_{\text{DL}}(\theta^\star(\phi); \mathcal{D}_{\text{RCT}})$ can be computed as:

$$
\begin{aligned}
\nabla_\phi \mathcal{L}_{\text{DL}}(\theta^\star(\phi); \mathcal{D}_{\text{RCT}}) &= \nabla_\theta \mathcal{L}_{\text{DL}}(\theta; \mathcal{D}_{\text{RCT}})|_{\theta = \theta^\star(\phi)} \cdot \frac{\partial \theta^\star(\phi)}{\partial \phi} \\
&= \nabla_\theta \mathcal{L}_{\text{DL}}(\theta; \mathcal{D}_{\text{RCT}})|_{\theta = \theta^\star(\phi)} \cdot (\nabla_\phi(-\alpha_\theta \nabla_\theta \mathcal{L}_{\text{PL}}(\phi, \theta; \mathcal{D}_{\text{OBS}}))) \\
&= -\alpha_\theta \cdot \nabla_\theta \mathcal{L}_{\text{DL}}(\theta; \mathcal{D}_{\text{RCT}})|_{\theta = \theta^\star(\phi)} \cdot \nabla_\phi \nabla_\theta \mathcal{L}_{\text{PL}}(\phi, \theta; \mathcal{D}_{\text{OBS}})
\end{aligned} \tag{23}
$$

Here, $\frac{\partial \theta^*(\phi)}{\partial \phi}$ is computed by differentiating through the single gradient descent update in Eq. (22):

$$\frac{\partial \theta^*(\phi)}{\partial \phi} = -\alpha_\theta \cdot \frac{\partial^2 \mathcal{L}_{\text{PL}}(\phi, \theta; \mathcal{D}_{\text{OBS}})}{\partial \phi \partial \theta}. \tag{24}$$

This approach, known as the Explicit Differentiation Algorithm, enables end-to-end optimization of the bridge model parameters $\phi$ using standard automatic differentiation frameworks. However,

the Explicit Differentiation Algorithm relies heavily on the optimization path and, when combined with decision loss, is often susceptible to vanishing gradients and suboptimal solutions. It should be emphasized that the assumption of reaching the optimum in one gradient descent step is often unrealistic. In practice, a single update typically leads to a suboptimal solution, whereas multiple updates can result in severe vanishing gradient issues.

### A.6 More details of the conjugate gradient (CG) Algorithm in Sec.4.3

In this appendix, we provide additional details on the conjugate gradient (CG) algorithm, which serves as a component within Algorithm 1 described in Sec. 4.3. Note that the conjugate gradient (CG) algorithm is employed to efficiently solve the following large-scale linear system:

$$\frac{\partial^2 \mathcal{L}_{\mathrm{PL}}(\phi, \theta; \mathcal{D}_{\mathrm{OBS}})}{\partial \theta^2}\bigg|_{\theta=\theta^\star(\phi)} \cdot \frac{\partial \theta^\star(\phi)}{\partial \phi} = - \frac{\partial^2 \mathcal{L}_{\mathrm{PL}}(\phi, \theta; \mathcal{D}_{\mathrm{OBS}})}{\partial \phi \partial \theta}\bigg|_{\theta=\theta^\star(\phi)}$$

which is the same as Eq. (18). The core idea of the CG algorithm is that solving $Ax = b$ is equivalent to minimizing the quadratic function $\frac{1}{2}x^\top A x - b^\top x$. Moreover, the CG algorithm can be implemented without explicit storage of large matrices by relying solely on matrix-vector products. For example, for a Hessian matrix $A = \nabla_\theta^2 \mathcal{L}$, the matrix-vector product $Ap$ can be computed as: $Ap = \nabla_\theta \left( p^\top \nabla_\theta \mathcal{L} \right)$, where $p$ is an arbitrary vector. This trick, which uses automatic differentiation twice, also applies to other matrices, enabling efficient implicit computation without explicit matrix construction. The pseudocode for the standard conjugate gradient algorithm is summarized in Algorithm 4.

---

**Algorithm 4** The Conjugate Gradient (CG) Algorithm

---

    **Input:** Matrix $A$; Vector $b$; Initial guess $x_0$; Tolerance $\epsilon$; Maximum iterations $n_{cg}$.

1:   $x \leftarrow x_0$                                                 ▷ Initialize solution
2:   $r \leftarrow b - Ax$                                     ▷ Compute initial residual
3:   $p \leftarrow r$                                             ▷ Set initial search direction
4:   **for** $k = 0, 1, 2, \ldots, n_{cg} - 1$ **do**
5:        **if** $\|r\| < \epsilon$ **then**
6:            **break**                                  ▷ Converged
7:        **end if**
8:        $\alpha \leftarrow \frac{r^\top r}{p^\top Ap}$                                ▷ Step size
9:        $x \leftarrow x + \alpha p$                            ▷ Update solution
10:       $r_{\mathrm{new}} \leftarrow r - \alpha Ap$                     ▷ Update residual
11:       **if** $\|r_{\mathrm{new}}\| < \epsilon$ **then**
12:           **break**                               ▷ Converged
13:       **end if**
14:       $\beta \leftarrow \frac{r_{\mathrm{new}}^\top r_{\mathrm{new}}}{r^\top r}$                       ▷ Update coefficient
15:       $p \leftarrow r_{\mathrm{new}} + \beta p$                ▷ Update search direction
16:       $r \leftarrow r_{\mathrm{new}}$                       ▷ Prepare for next iteration
17: **end for**

    **Output:** Solution $x$ such that $Ax \approx b$

---

### A.7 Dual Decision Loss ($\mathcal{L}_{\mathrm{DDL}}$), Dual Policy Learning Loss ($\mathcal{L}_{\mathrm{DPL}}$), and Dual Improved Finite Difference Strategy (DIFD)

We provide an alternative formulation of the decision loss, termed the dual decision loss, which is designed to directly quantify the decision quality of the dual formulations of MTBAP.

$$\mathcal{L}_{\mathrm{DDL}}(\theta) = -M \cdot \mathbb{E}_{i \in [N], j \in [M]} \sum_\lambda \left[ z_{ij}^{\mathrm{dual}}(\hat{r}(\theta), \hat{c}(\theta)) \cdot (r_{ij} - \lambda \cdot c_{ij}) \right]. \tag{25}$$

$\mathcal{L}_{\mathrm{DDL}}$ is also non-computed due to the lack of the counterfactual responses. By leveraging strong ignorability $(X, R(t), C(t)) \perp T$ of experimental data, we derive an unbiased estimator of the dual decision loss as follows:

$$\mathcal{L}_{\mathrm{DDL}}(\theta; \mathcal{D}_{\mathrm{RCT}}) = -\mathbb{E}_{i, t_i} \sum_\lambda \left[ \frac{N}{N_{t_i}} \cdot z_{it_i}^{\mathrm{dual}}(\hat{r}(\theta), \hat{c}(\theta)) \cdot (r_{it_i} - \lambda \cdot c_{it_i}) \right]. \tag{26}$$

$N_{t_i}$ is the number of individuals assigned treatment $t_i$ in $\mathcal{D}_{\mathrm{RCT}}$. Note that in this context, $\lambda$ is not the optimal Lagrange multiplier $\lambda^*$ obtained by binary search, but rather a user-specified hyperparameter. It represents a discrete interpolation over arbitrary budget constraints $B$.

$\mathcal{L}_{\mathrm{DDL}}$ is continuously differentiable with respect to $z^*_{it_i}(\hat{r}(\theta), \hat{c}(\theta))$. Based on the Lagrangian relaxation algorithm $\mathcal{A}$ (1), and given arbitrary $\lambda$, the solution is:

$$z^{\mathrm{dual}}_{it_i}(\hat{r}(\theta), \hat{c}(\theta)) = \mathbb{1}\left\{ t_i = \arg\max_{j \in [M]} [\hat{r}_{ij}(\theta) - \lambda \hat{c}_{ij}(\theta)] \right\} \tag{27}$$

where $\mathbb{1}$ is indicator function and $\lambda$ is an arbitrary user-specified Lagrange multiplier. Due to the existence of indicator functions, $z^*_{it_i}(\hat{r}(\theta), \hat{c}(\theta))$ is non-differentiable with respect to $\theta$. By utilizing Softmax functions, the discrete solution $z^{\mathrm{dual}}_{it_i}(\hat{r}(\theta), \hat{c}(\theta))$ can be relaxed to a continuously differentiable function $z^{\mathrm{dual}'}_{it_i}(\hat{r}(\theta), \hat{c}(\theta))$, which can also be regarded as the probability of $z^{\mathrm{dual}}_{it_i} = 1$:

$$z^{\mathrm{dual}'}_{it_i}(\hat{r}(\theta), \hat{c}(\theta)) = \frac{\exp[\hat{r}_{it_i}(\theta) - \lambda \hat{c}_{it_i}(\theta)]}{\sum_{j \in [M]} \exp[\hat{r}_{ij}(\theta) - \lambda \hat{c}_{ij}(\theta)]}, \tag{28}$$

Hence, we obtain a surrogate decision loss $\mathcal{L}_{\mathrm{DPL}}$ of $\mathcal{L}_{\mathrm{DDL}}$, called the dual policy learning loss:

$$\mathcal{L}_{\mathrm{DPL}}(\theta; \mathcal{D}_{\mathrm{RCT}}) = -\mathbb{E}_{i, t_i} \sum_{\lambda} \left[ \frac{N}{N_{t_i}} \cdot \frac{\exp[\hat{r}_{it_i}(\theta) - \lambda \hat{c}_{it_i}(\theta)]}{\sum_{j \in [M]} \exp[\hat{r}_{ij}(\theta) - \lambda \hat{c}_{ij}(\theta)]} \cdot (r_{it_i} - \lambda \cdot c_{it_i}) \right], \tag{29}$$

While the dual loss considers all budget levels, our proposed $\mathcal{L}_{\mathrm{PPL}}$ in the main text directly targets decision quality under a specific budget $B$, thereby better aligning with real-world marketing constraints. We also introduce the Dual Improved Finite Difference (DIFD) strategy, which estimates the gradients $\frac{\partial \mathcal{L}_{\mathrm{DDL}}(\theta; \mathcal{D}_{\mathrm{RCT}})}{\partial z^{\mathrm{dual}'}_{ij}(\hat{r}(\theta), \hat{c}(\theta))}$ via black-box perturbations on $\mathcal{L}_{\mathrm{DDL}}$, and accelerates computation using a $\mathcal{L}_{\mathrm{DPL}}$-aware gradient estimator. Compared to $\mathcal{L}_{\mathrm{DPL}}$, DIFD preserves the dual optimization landscape without relaxation, and, by freezing the computed gradients as non-trainable nodes, enables seamless integration with automatic differentiation libraries. The surrogate decision loss $\mathcal{L}_{\mathrm{DIFD}}$ is given by:

$$\mathcal{L}_{\mathrm{DIFD}}(\theta; \mathcal{D}_{\mathrm{RCT}}) = \mathbb{E}_{i \in [N], j \in [M]} \sum_{\lambda} \left[ \frac{\partial \mathcal{L}_{\mathrm{DDL}}(\theta; \mathcal{D}_{\mathrm{RCT}})}{\partial z^{\mathrm{dual}'}_{ij}(\hat{r}(\theta), \hat{c}(\theta))} \cdot z^{\mathrm{dual}'}_{ij}(\hat{r}(\theta), \hat{c}(\theta)) \right]. \tag{30}$$

Also, the pseudocode for the $\mathcal{L}_{\mathrm{DPL}}$-aware gradient estimator in DIFD is provided in Algorithm 5.

# B  More Details of Offline Experiment

Here, we provide additional details for Sec.5.1 (Offline Experimental Setup) and Sec.5.2 (Offline Experimental Results) of the main text.

## B.1  Details of Offline Experimental Setup

Here, we provide additional information for Sec. 5.1 (Offline Experimental Setup) of the main text, covering the dataset and preprocessing, evaluation metrics, and experimental details.

### B.1.1  CRITEO-UPLIFT v2 (Hybrid)

**CRITEO-UPLIFT v2.** This public dataset is provided by the AdTech company Criteo in the AdKDD'18 workshop[11]. The dataset contains 13.9 million samples collected from a random control trial (RCT) that prevents a random part of users from being targeted by advertising. Each sample has 12 features, 1 binary treatment indicator and 2 response labels(visit/conversion). In order to study resource allocation problem under limited budget using the dataset, we follow[51] and take the visit/conversion label as the cost/value respectively. To better reflect real-world marketing scenarios where OBS data far outnumbers RCT data, we simulate a marketing policy to convert part of RCT data into OBS data. We refer to this as CRITEO-UPLIFT v2 (Hybrid).

**CRITEO-UPLIFT v2 (Hybrid).** Given a total of 13.9 million RCT samples, we use 5% of the data to train a two-stage model with the standard cross-entropy loss. This trained model is then used to

**Algorithm 5** $\mathcal{L}_{\text{DPL}}$-aware gradient estimator of the dual improved finite difference strategy (DIFD)

---

**Input:** Lagrange multiplier $\lambda$; data $D \equiv \{(x_i, t_i, r_{it_i}, c_{it_i})\}_{i=1}^N$; predicted revenue/cost $\hat{r}, \hat{c}$.
**Compute:** For each $t$ in $[M]$, $N_t \leftarrow$ number of samples with $t_i = t$; $p_t \leftarrow N_t/N$.
**Initialize:** $\frac{\partial \mathcal{L}_{\text{DL}}(\theta; \mathcal{D}_{\text{RCT}})}{\partial z_{ij}^{\text{dual}'}(\hat{r}(\theta), \hat{c}(\theta))} = 0$, $z_{ij} = 0$ for all $i, j$.

1: $\forall i, j$, $a_{ij} = (r_{ij} - \lambda \cdot c_{ij})$, $z_{ij} = \mathbb{I}_{j = \arg\max_j (r_{ij} - \lambda \cdot c_{ij})}$
2: $\bar{r}(\lambda, r, c, \hat{r}, \hat{c}) \leftarrow \frac{1}{N} \sum_i \frac{1}{p_{t_i}} r_{t_i} \mathbb{I}_{t_i = \arg\max_j z_{ij}}$
3: $\bar{c}(\lambda, r, c, \hat{r}, \hat{c}) \leftarrow \frac{1}{N} \sum_i \frac{1}{p_{t_i}} c_{t_i} \mathbb{I}_{t_i = \arg\max_j z_{ij}}$
4: $-\mathcal{L}_{DDL}(\lambda, r, c, \hat{r}, \hat{c}) \leftarrow \bar{r}(\lambda, r, c, \hat{r}, \hat{c}) - \lambda \cdot \bar{c}(\lambda, r, c, \hat{r}, \hat{c})$
5: matching_indices $= \{i \mid t_i = \arg\max_j z_{ij}, \forall i\}$
6: mismatching_indices $= \{i \mid t_i \neq \arg\max_j z_{ij}, \forall i\}$
7: **for all** $i \in$ matching_indices **do**
8: $\quad h_{it_i}^z = \max_{j \neq t_i} a_{ij} - a_{it_i}$
9: $\quad \frac{\partial -\mathcal{L}_{DDL}}{\partial z_{it_i}^{\text{dual}'}} = \frac{-\frac{1}{N \cdot p_{t_i}} \cdot (r_{it_i} - \lambda \cdot c_{it_i})}{h_{it_i}^z}$
10: $\quad$ **for all** $j \in \{1, 2, ..., M\}$, $j \neq t_i$ **do**
11: $\quad\quad h_{ij}^z = a_{it_i} - a_{ij}$
12: $\quad\quad \frac{\partial -\mathcal{L}_{DDL}}{\partial z_{ij}^{\text{dual}'}} = \frac{-\frac{1}{N \cdot p_{t_i}} \cdot (r_{it_i} - \lambda \cdot c_{it_i})}{h_{ij}^z}$
13: $\quad$ **end for**
14: **end for**
15: **for all** $i \in$ mismatching_indices **do**
16: $\quad j = \arg\max_j a_{ij}$
17: $\quad h_{it_i}^z = a_{ij} - a_{it_i}$, $h_{ij}^z = -h_{it_i}^r$
18: $\quad \frac{\partial -\mathcal{L}_{DDL}}{\partial z_{it_i}^{\text{dual}'}} = \frac{\frac{1}{N \cdot p_{t_i}} \cdot (r_{it_i} - \lambda \cdot c_{it_i})}{h_{it_i}^z}$
19: $\quad \frac{\partial -\mathcal{L}_{DDL}}{\partial z_{ij}^{\text{dual}'}} = \frac{\frac{1}{N \cdot p_{t_i}} \cdot (r_{it_i} - \lambda \cdot c_{it_i})}{h_{ij}^z}$
20: **end for**
**Output:** the gradients $\frac{\partial \mathcal{L}_{\text{DDL}}(\theta; \mathcal{D}_{\text{RCT}})}{\partial z_{ij}^{\text{dual}'}(\hat{r}(\theta), \hat{c}(\theta))} = -\frac{\partial -\mathcal{L}_{\text{DDL}}(\theta; \mathcal{D}_{\text{RCT}})}{\partial z_{ij}^{\text{dual}'}(\hat{r}(\theta), \hat{c}(\theta))}$.

---

simulate a marketing policy on 50% of the total RCT samples. We construct the observational (OBS) dataset by selecting users for whom the coupon assignment under the simulated policy matches the actual assignment in the data. Note that this procedure discards unmatched RCT samples, resulting in an OBS dataset with 3,498,294 samples, which accounts for approximately 25% of the total data. Our analysis shows that the constructed OBS dataset achieves an 82.43% improvement in ROI compared to the random dataset, which demonstrates that the constructed OBS dataset closely reflects observational data generated by real-world marketing strategies. Excluding the 55% of random data that is not utilized in the above process, we further split the remaining 45% RCT samples into 5% for the RCT training set, 10% for validation set, and 30% for test set. To summarize, the resulting datasets contain 3,498,294 samples in the OBS training set, 698,960 in the RCT training set, 1,397,959 in the RCT validation set, and 4,193,878 in the RCT test set. It is worth noting that the ratio of OBS to RCT samples in the training set is approximately 5:1.

### B.1.2 EOM (Expected Outcome Metric).

**EOM (Expected Outcome Metric).** EOM is a common metric for marketing optimization in [2, 51, 49, 52]. Based on RCT data, an unbiased estimation of the expected outcome (per-capita revenue/per-capita cost) for arbitrary budget allocation policy can be obtained. Since EOM represents the decision quality of marketing under multilple treatments, we use EOM to compare the performance of different methods in Marketing data I and II. We give pseudocode of EOM (Expected Outcome Metric) for unbiased estimation of per-capita revenue or cost in Algorithm 6.

---

**Algorithm 6** EOM: Unbiased estimation of expected outcome (per-capita revenue or cost) for Lagrangian budget allocation policy $\mathcal{A}$ with predicted revenue $\hat{r}$ and cost $\hat{c}$ under budget $B$.

---

**Input:** data $D = \{(x_i, t_i, r_{it_i}, c_{it_i})\}_{i=1}^N$; budget $B$; predicted revenue/cost $\hat{r}, \hat{c}$; small constant $\epsilon$

**Compute:** For each $t$ in $[M]$, $N_t \leftarrow$ number of samples with $t_i = t$; $p_t \leftarrow N_t/N$.

**Initialize:** $\lambda_{\min} \leftarrow 0$, $\lambda_{\max} \leftarrow \max_{i,j}\left(\frac{\hat{r}_{ij}}{\hat{c}_{ij}}\right)$, $z_{ij} \leftarrow 0$ for all $i, j$

1: **while** $\lambda_{\max} - \lambda_{\min} > \epsilon$ **do**
2:      $\lambda \leftarrow \frac{\lambda_{\max} + \lambda_{\min}}{2}$
3:      **for all** $i, j$ **do**
4:          $z_{ij} \leftarrow \mathbb{I}\left(j = \arg\max_j(\hat{r}_{ij} - \lambda\hat{c}_{ij})\right)$
5:      **end for**
6:      $\bar{r}(\lambda, r, c, \hat{r}, \hat{c}) \leftarrow \frac{1}{N}\sum_i \frac{1}{p_{t_i}} r_{t_i} \mathbb{I}\left(t_i = \arg\max_j z_{ij}\right)$
7:      $\bar{c}(\lambda, r, c, \hat{r}, \hat{c}) \leftarrow \frac{1}{N}\sum_i \frac{1}{p_{t_i}} c_{t_i} \mathbb{I}\left(t_i = \arg\max_j z_{ij}\right)$
8:      **if** $\left|\frac{B}{N} - \bar{c}(\lambda, r, c, \hat{r}, \hat{c})\right| < \epsilon$ **then**
9:          **break**
10:     **end if**
11:     **if** $\frac{B}{N} - \bar{c}(\lambda, r, c, \hat{r}, \hat{c}) > 0$ **then**
12:        $\lambda_{\max} \leftarrow \lambda$
13:     **else**
14:        $\lambda_{\min} \leftarrow \lambda$
15:     **end if**
16: **end while**
17: $\lambda^* \leftarrow \lambda$
18: $\bar{r}(B, r, c, \hat{r}, \hat{c}) \leftarrow \bar{r}(\lambda^*, r, c, \hat{r}, \hat{c})$
19: $\bar{c}(B, r, c, \hat{r}, \hat{c}) \leftarrow \bar{c}(\lambda^*, r, c, \hat{r}, \hat{c})$
     **Output:** expected per capita revenue $\bar{r}(B, r, c, \hat{r}, \hat{c})$, expected per capita cost $\bar{c}(B, r, c, \hat{r}, \hat{c})$, $\lambda^*$;

---

### B.1.3 Experimental Details

**Model Architecture.** For CRITEO-UPLIFT v2 (Hybrid), we employ a 4-layer multi-head multilayer perceptron (MLP) with layer sizes of 64-32-32-4, where the first two outputs correspond to predicted revenue and the remaining outputs correspond to predicted cost. For Marketing Data I, we use a 4-layer multi-head MLP with layer sizes of 128-64-32-16; in this case, the first eight outputs represent predicted revenue, and the remaining outputs represent predicted cost. For Marketing Data II, the model is a 4-layer multi-head MLP with layer sizes of 128-64-32-10, where the first five outputs are for predicted revenue and the remaining outputs are for predicted cost. Note that, unless otherwise specified, the target model, bridge model, and teacher model all adopt the same architecture.

**Device.** All experiments are conducted on two NVIDIA A100 GPUs with a total of 232 GB memory.

**Optimizer.** We use the Adam optimizer for training.

**Training Procedure.** In the three experiments, the models are trained for 100, 500, and 500 epochs, respectively. For each experiment, the model checkpoint with the highest AUCC/EOM on the validation set is selected as the best model.

**Other Hyperparameters.** The number of gradient descent (GD) steps for assumed updates, $k$, is set to 5. The number of conjugate gradient iterations, $n_{\text{cg}}$, is set to 50. The warm-start period for Bi-DFCL, if applicable, is set to 20 epochs.

### B.2 Details of Ablation Studies.

To show the effects of individual components, we conduct ablation study by incrementally adding four key components of Bi-DFCL to baseline in a sequential manner: Decision Loss (PPL), Bi-level Optimization by hybrid RCT and OBS data, Counterfactual Labels, and Implicit Differentiation Algorithm. The experimental results on marketing datasets are reported in Table 3 or Table 5.

Specifically, the baselines corresponding to each row in Table 3 are described as follows:

Table 5: Ablation study of each individual component in Bi-DFCL with two marketing datasets.

| Components of Bi-DFCL | | | | Marketing Data I | | Marketing Data II | |
|---|---|---|---|---|---|---|---|
| Decision Loss (PPL) | Bi-level Optimization | Counterfactual Labels | Implicit Differentiation | EOM | Improvement | EOM | Improvement |
| ✗ | ✗ | ✗ | ✗ | 1.0000 | – | 1.0000 | – |
| ✓ | ✗ | ✗ | ✗ | 1.0167 | 1.67% | 1.0156 | 1.56% |
| ✓ | ✓ | ✗ | ✗ | 1.0240 | 2.40% | 1.0175 | 1.75% |
| ✓ | ✓ | ✓ | ✗ | 1.0248 | 2.48% | 1.0213 | 2.13% |
| ✓ | ✓ | ✓ | ✓ | **1.0277** | **2.77%** | **1.0252** | **2.52%** |

**Row 1 (Baseline):** This is the TSM-SL baseline trained on RCT data only, without any of the proposed components. It serves as the basic reference model.

**Row 2 (Baseline + Decision Loss):** This variant corresponds to DFCL-PPL, which incorporates the decision loss ($\mathcal{L}_{\mathrm{PPL}}$) on RCT data, but does not include bi-level optimization.

**Row 3 (Baseline + Decision Loss + Bi-level Optimization):** This setting corresponds to Bi-DFCL-PPL without using synthesized counterfactual pseudo-labels to parameterize $\mathcal{L}_{\mathrm{PL}}$. Instead, it employs an improved version of IPW (inverse propensity weighting), where the bridge model directly outputs dynamically adaptive weights for reweighting factual samples, rather than using fixed or estimated propensity scores. Implicit differentiation is not employed here(i.e., explicit differentiation is used).

**Row 4 (Baseline + Decision Loss + Bi-level Optimization + Counterfactual Labels):** This configuration is Bi-DFCL-PPL, where synthesized counterfactual pseudo-labels are used to parameterize the $\mathcal{L}_{\mathrm{DPL}}$, but implicit differentiation is still not applied (i.e.,explicit differentiation is used).

**Row 5 (Full Model):** This is the complete Bi-DFCL-PPL with all four components enabled: decision loss (PPL), bi-level optimization, counterfactual labels, and implicit differentiation.

We can find that after the introduction of each module, the performance can all be strengthened to some extent, which demonstrates that our three contributions can all benefit the marketing optimization.

## B.3 Details of In-depth Analysis

**The effect of RCT and OBS training data size.** We first conduct an in-depth analysis to investigate the effect of training data size on performance using Marketing Data I, as well as to validate the bias-variance properties of RCT and OBS data. The experimental results are summarized in Table 6.

Table 6: Effect of training data size (OBS and RCT) on performance with Marketing Data I.

| Method | OBS | RCT | OBS:RCT Ratio | EOM | Improvement |
|---|---|---|---|---|---|
| TSM-SL | 2,220,781 | 0 | – | 0.9869 | -1.31% |
| TSM-SL | 0 | 2,220,781 | – | 1.0000 | – |
| TSM-SL | 22,201,405 | 0 | – | 1.0067 | 0.67% |
| Bi-DFCL-PPL | 22,201,405 | 222,000 | 100.01:1 | 1.0190 | 1.90% |
| Bi-DFCL-PPL | 22,201,405 | 1,100,000 | 20.18:1 | 1.0258 | 2.58% |
| Bi-DFCL-PPL | 22,201,405 | 2,220,781 | 10.00:1 | **1.0277** | **2.77%** |

As shown in Table 6, models trained solely on RCT data (e.g., TSM-SL with 2,220,781 RCT samples) serve as an unbiased reference, but their performance is limited by high variance due to the relatively small sample size. In contrast, models trained only on large-scale OBS data may suffer from bias, as reflected in lower EOM values when using 2,017,450 or even 3,381,5274 OBS samples alone. Notably, as the amount of OBS data increases from 2,220,781 to 22,201,405, the EOM improves from 0.9869 to 1.0067, which highlights that the low-variance property of large-scale observational data is highly beneficial for robust and high-quality decision making. Furthermore, when a sufficient amount of RCT data is combined with abundant OBS data (e.g., Bi-DFCL-PPL with 22,201,405 OBS and 2,220,781 RCT samples), the model achieves the best performance (EOM = 1.0277, Improvement = 2.77%). This demonstrates the effectiveness of leveraging large-scale observational data to reduce variance, together with a moderate amount of randomized data to correct for bias, thereby achieving a favorable bias-variance trade-off and superior overall model performance.

**The sensitivity of key hyperparameters.** We further evaluate the sensitivity of key hyperparameters, specifically the number of gradient descent (GD) steps for assumed updates ($k$, default = 5) and the number of conjugate gradient iterations ($n_{\mathrm{cg}}$, default = 50), by varying their values. The results on Marketing Data II are summarized in Table 7.

Table 7: Sensitivity analysis of key hyperparameters on performance with Marketing Data II.

| $k$ (GD Steps) | $n_{\mathrm{cg}}$ (CG Iterations) | EOM | Improvement |
|---|---|---|---|
| 1 | 10 | 1.0199 | 1.99% |
| 1 | 50 | 1.0217 | 2.17% |
| 5 | 10 | 1.0230 | 2.30% |
| 5 | 50 | 1.0252 | 2.52% |
| 5 | 100 | 1.0253 | 2.53% |
| 5 | 200 | 1.0249 | 2.49% |
| 10 | 50 | 1.0255 | 2.55% |
| 10 | 100 | 1.0252 | 2.52% |

As shown in Table 7, the performance of our method is relatively stable across a range of values for $k$ and $n_{\mathrm{cg}}$, indicating that the proposed approach is robust to these hyperparameter settings. Notably, when $k = 1$, the implicit differentiation algorithm does not provide a significant advantage over explicit differentiation. This suggests that the strength of implicit differentiation lies in its independence from the optimization path, allowing for any number of iterative updates to reach the optimal solution, rather than relying on the overly strong assumption of explicit differentiation that a single gradient descent step suffices to achieve optimality.

**The Robustness of Bi-DFCL.** Moreover, we evaluate the robustness of Bi-DFCL under multiple sets of budget values $B$. The results on Marketing Data I and II are summarized in Figure 3.

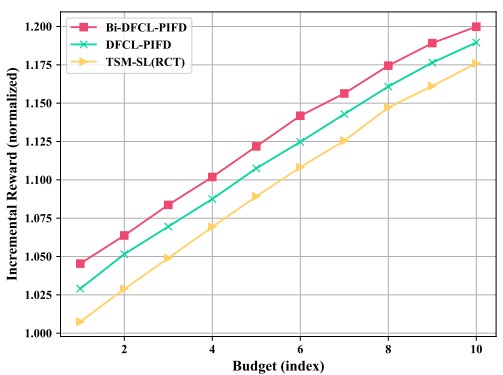 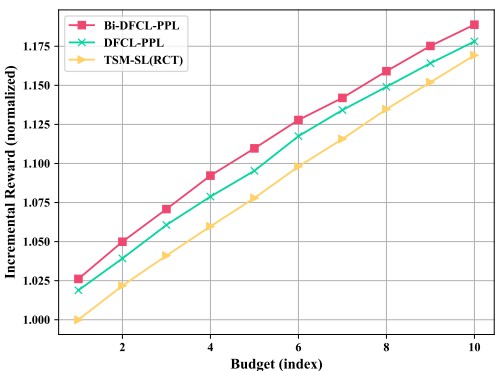

(a) Incremental reward (normalized EOM) on Marketing Data I across 10 budget levels.

(b) Incremental reward (normalized EOM) on Marketing Data II across 10 budget levels.

Figure 3: Robustness of Bi-DFCL under multiple budget values $B$ on Marketing Data I and II.

As illustrated in Figure 3, Bi-DFCL consistently achieves higher incremental reward (EOM) across a range of candidate budget values on both Marketing Data I and II. This demonstrates the robustness and effectiveness of Bi-DFCL in maintaining superior decision quality when the budget varies within several candidate levels, further highlighting its practical applicability in real-world marketing.

**Computational Efficiency Analysis.** Finally, we provide a comprehensive analysis of the computational overhead of Bi-DFCL from both space and time efficiency perspectives.

**Space Efficiency:** Bi-DFCL does not incur additional space overhead compared to existing baselines. While implicit differentiation algorithms typically require storing large-scale inverse matrices, we employ the Conjugate Gradient (CG) algorithm to avoid this issue. The CG algorithm circumvents the storage of large-scale inverse matrices through matrix-vector products (see Appendix A.6).

**Time Efficiency:** For online inference, Bi-DFCL only uses the well-trained target model, resulting in inference time identical to simple causal learning methods. However, additional time overhead occurs during offline training. Table 8 compares the training time of different methods on Marketing Data II.

For fairness, all methods were fully trained for 500 epochs using the same model structure (no early stopping). As shown in Table 8, Bi-DFCL requires approximately 6-7 times the training time of the simplest causal method TSM-SL. The ablation studies reveal that most time overhead stems from solving the bi-level optimization problem. Our use of implicit differentiation with the CG algorithm

Table 8: Comprehensive analysis of training time (minutes) across different methods

| Method | Data | Training Time (min) | Relative to TSM-SL |
|---|---|---|---|
| TSM-SL | RCT | 2.505 | 0.06× |
| DFCL-PPL | RCT | 3.163 | 0.07× |
| DFCL-PIFD | RCT | 9.948 | 0.23× |
| TSM-SL | OBS | 39.918 | 0.94× |
| TSM-SL | RCT+OBS | 42.332 | 1.00× |
| KD-Label | RCT+OBS | 67.358 | 1.59× |
| LTD-DR | RCT+OBS | 492.559 | 11.63× |
| AutoDebias | RCT+OBS | 397.886 | 9.40× |
| Bi-DFCL-PPL | RCT+OBS | 265.263 | 6.26× |
| Bi-DFCL-PIFD | RCT+OBS | 294.927 | 6.96× |
| Bi-DFCL-PPL w/o ID | RCT+OBS | 345.132 | 8.15× |
| Bi-DFCL-PIFD w/o ID | RCT+OBS | 427.515 | 10.10× |

provides two key advantages: (1) it reduces time complexity from $O(n^3)$ to $O(n)$ by avoiding matrix inversion, and (2) it obtains more accurate optimal solutions, allowing bilevel optimization solving once every $k$ batches rather than every batch. The comparison between Bi-DFCL variants with and without implicit differentiation (ID) demonstrates the efficiency gains of our approach. In summary, although Bi-DFCL introduces additional offline training time, this investment is justified by significantly improved online decision quality. Our further improvements also effectively mitigate this overhead, making Bi-DFCL practical for real-world marketing applications.

## C    Boarder Impacts

Our work offers several positive societal impacts. First, by improving the decision quality of marketing resource allocation, our method helps platforms maximize the effectiveness of their marketing campaigns under real-world budget constraints. This can lead to increased user engagement and satisfaction, as users are more likely to receive relevant and timely offers. Second, the reduction of resource waste contributes to more sustainable business operations, which benefits both companies and consumers. Third, our approach has demonstrated strong performance in both offline benchmarks and large-scale online deployments, indicating its practical value for the digital economy. The adoption of such data-driven decision-making tools can further support innovation and the healthy development of the broader digital marketing ecosystem.

