# OpenReview forum: "Bi-Level Decision-Focused Causal Learning for Large-Scale Marketing Optimization: Bridging Observational and Experimental Data"
_NeurIPS.cc/2025/Conference — NeurIPS 2025 poster_

### Official Review · Reviewer_HrqG · 2025-07-02

**Clarity:** 3
**Significance:** 3
**Originality:** 3
**Rating:** 4
**Confidence:** 3

**Summary:**

The paper proposes Bi-Level Decision-Focused Causal Learning (Bi-DFCL), a novel framework for large-scale marketing optimization that bridges observational and experimental data. It addresses two key challenges: (1) prediction-decision misalignment in traditional two-stage methods and (2) bias-variance tradeoffs between biased but abundant OBS data and unbiased but scarce RCT data. Bi-DFCL introduces a bi-level optimization framework where an upper-level unbiased OR estimator dynamically corrects the lower-level ML model via a Bridge Network. The method demonstrates significant improvements over baselines in offline benchmarks and real-world A/B tests on a major food delivery platform.

**Questions:**

- The bi-level optimization and implicit differentiation are elegant but computationally intensive. Could you provide runtime comparisons across datasets? How does Bi-DFCL scale with the number of treatments or sample size?
- For large-scale deployments, have you explored approximations to reduce overhead?
- Table 3 suggests 'Implicit Differentiation' contributes marginally to gains (e.g., +0.29% in Marketing Data I). Is this due to optimization stability, or are there cases where it's critical?

**Ethical Concerns:**

["NO or VERY MINOR ethics concerns only"]

**Final Justification:**

I keep my original justification, weak accept.

**Limitations:**

yes

**Quality:**

3

**Strengths And Weaknesses:**

Strengths
- The bi-level optimization design bridges OBS and RCT data, leveraging their complementary strengths. The Bridge Network and implicit differentiation algorithm are novel contributions.
- Extensive validation on public benchmarks, industrial datasets, and online A/B tests underscores real-world applicability.
- The paper also provides theoretical grounding, i.e., unbiased estimator proofs.
- The paper is well-written with clear and logical structure.

Weaknesses
- The bi-level optimization and implicit differentiation may incur high computational costs for large datasets. Scalability details (e.g., runtime comparisons) are not thoroughly discussed.
- The focus on marketing raises questions about applicability to other domains (e.g., healthcare, logistics). The method’s reliance on RCT data might limit adoption in settings where RCTs are infeasible.
- Table 3’s incremental ablation could better isolate the impact of counterfactual labels and implicit differentiation. For instance, how much does the Bridge Network alone contribute vs. the full bi-level setup?

---

> ### Author Rebuttal · Authors · 2025-07-30
>
> We sincerely appreciate your approval of the idea and the novelty of this work, and thank you for the helpful suggestions. Below, we hope to address your concerns and questions to improve the clarity and readability of our paper.
>
> ---
>
> ### **【Q1】The  computational overhead**
>
> **[Q1]: ”Could you provide runtime comparisons across datasets? How does Bi-DFCL scale with the number of treatments or sample size?”**
>
> **[Q1Response]:**
>
> * **Online Inference:**
>
> 	* We first want to clarify that for the online deployment and inference of Bi-DFCL, we only use the well-trained Target Model (see the Output of Algorithm1).
> 	* Therefore, **the inference time of our method is exactly the same as that of simple causal learning methods—because their model structures are identical**.
>
> * **Offline Training:**
>
> 	* However, Bi-DFCL does introduce additional time overhead during offline training—which is one of the issues we have been working to address.
>
> 	1. **Runtime Comparisons with different baselines**
>
> 		  * As suggested by the reviewer, we compare the training time (minutes) of different baselines (using RCT + OBS data for training) on the Marketing data II. The results are shown below.
>
> 		* | Dataset II |     Methods      | Training time |
> 			| :--------: | :--------------: | :-----------: |
> 			|  RCT+OBS   |      TSM-SL      |    42.332     |
> 			|  RCT+OBS   |     KD-Label     |    67.358     |
> 			|  RCT+OBS   |      LTD-DR      |    492.559    |
> 			|  RCT+OBS   |    AutoDebias    |    397.886    |
> 			|  RCT+OBS   | **Bi-DFCL-PPL**  |  **265.263**  |
> 			|  RCT+OBS   | **Bi-DFCL-PIFD** |  **294.927**  |
>
>
> 		  * Note: For fairness, all methods were fully trained for 500 epochs using the same model structure (no early stopping).
>
> 	2. **Runtime scale with the number of treatments or sample size**
>
> 		  * As suggested by the reviewer, we also compare the training time (minutes)  scale with the number of sample size. The results are shown below.
>
> 		* | Dataset II (RCT+OBS) |   Methods    | Training time |
> 			| :------------------: | :----------: | :-----------: |
> 			|         100%         |    TSM-SL    |    42.332     |
> 			|         50%          |    TSM-SL    |    20.207     |
> 			|         25%          |    TSM-SL    |    10.738     |
> 			|         100%         | Bi-DFCL-PPL  |    265.263    |
> 			|         50%          | Bi-DFCL-PPL  |    124.993    |
> 			|         25%          | Bi-DFCL-PPL  |    59.193     |
> 			|         100%         | Bi-DFCL-PIFD |    294.927    |
> 			|         50%          | Bi-DFCL-PIFD |    125.391    |
> 			|         25%          | Bi-DFCL-PIFD |    66.839     |
>
> 		  * From the table, the following conclusions can be drawn:
>
> 			   * Our proposed Bi-DFCL, like the baseline TSM-SL, shows a roughly **linear increase in training time as the sample size changes**.
>
> 			   * As a supplement, **the training time does not increase with the number of treatments** (the minimal increase in time is negligible).
>
> 		  * In addition, we **theoretically explain** why Bi-DFCL can achieve a linear increase:
>
> 			* This is because most of the training time of Bi-DFCL comes from solving the bi-level optimization problem - implicit differentiation.
>
> 			* Implicit differentiation itself requires calculating the inverse of a large-scale matrix, which theoretically has a time complexity of O(n³)  (where n is the number of model parameters).
>
> 			* However, we use a numerical method — the CG algorithm — to solve this problem, reducing the time complexity of this step from O(n³) to O(n).
>
> 	3. **Summary**
>
> 		* Bi-DFCL does not incur additional space overhead or inference time overhead, but it **does have a training time overhead 6-7 times that of the baseline TSM-SL**.
> 		* **Bi-DFCL achieve a linear increase in training time with the number of samples.**
> 		* However, on the one hand, the extra **offline training time is absolutely worthwhile in exchange for higher online decision quality**. On the other hand, we have **also made several improvements to address this issue (such as the CG algorithm)**.
>
> ---
>
> ### **【Q2】Approximations to reduce overhead**
>
> **[Q2]: ”For large-scale deployments, have you explored approximations to reduce overhead?”**
>
> **[Q2Response]:**
>
> * Yes, we have tried many ways to reduce the training time overhead of Bi-DFCL — this is one of our key goals.
> * **Online Inference (large-scale deployments):**  the inference time of our method is exactly the same as that of simple causal learning methods.
> * **Offline Training:** The following are the methods we ultimately adopted that can effectively reduce the training time overhead：
>
> 	1. **Implicit differentiation and CG algorithm for solving the BLO problems**:
> 		* The CG algorithm efficiently solves the following large-scale linear system through numerical methods, thereby avoiding matrix inversion — reducing the time complexity from O(n³) to O(n).
>
> 	2. **Hyperparameter—*k* (number of GD steps for assumed updates)**:
>
> 		* Implicit differentiation, as it does not depend on the optimization path, can obtain a more accurate optimal solution compared to explicit differentiation(assuming one step GD to reach optimal). This allows solving the bi-level optimization problem once every k batches, rather than having k=1 as in explicit differentiation.
>
> 		* We use k=5 by default, which eliminates the need to solve the BLO problem every time, thereby significantly reducing the time overhead.
>
> 	3. **Gating coefficients, warm starts to accelerate convergence:**
>
> 		* We have done a lot of work to reduce the training variance of Bi-DFCL — for example, gating coefficients are used to generate more stable counterfactual pseudo-labels, and warm starts are used to accelerate convergence.
>
> 		* These efforts are aimed at training stability, which can greatly accelerate the convergence of BLO problems, thereby reducing time overhead.
>
> ---
>
> ### **【Q3】Contributions of Implicit Differentiation**
>
> **【Q3】: ”Table 3 suggests 'Implicit Differentiation' contributes marginally to gains (e.g., +0.29% in Marketing Data I). Is this due to optimization stability, or are there cases where it's critical?”**
>
> **[Q3Response]:**
>
> * Yes, the advantages of Implicit Differentiation (and the CG algorithm used therein) mainly include three aspects: **more accurate gradients; more stable gradients; and higher solution efficiency**.
> * The contributions of Implicit Differentiation shown in Table 3 (such as 0.29% and 0.39%) stem from its first advantage: more accurate gradients.
> * However, **the other two advantages are also very important (especially gradient stability)**.
> * The commonly used existing solution is the explicit differentiation algorithm(see Appendix A.4 for details). We summarize the advantages of Implicit Differentiation as follows:
> 	* **more accurate gradients**：
> 		* Explicit Differentiation Algorithm relies heavily on the optimization path and, when combined with decision loss, is often susceptible to vanishing gradients and suboptimal solutions.
> 		* It should be emphasized that the assumption of reaching the optimum in one gradient descent step is often unrealistic. In practice, a single update typically leads to a suboptimal solution, whereas multiple updates can result in severe vanishing gradient issues.
> 	* **more stable gradients**：
> 		* As introduced above, the vanishing gradient problem often occurs in the explicit differentiation algorithm.
> 		* In practice, we found that its training process is very unstable, so it is necessary to print gradient information for checking every update, and the experimental results are the optimal results when the vanishing gradient problem is not serious.
> 		* That is to say, the results of our implicit differentiation algorithm are more stable with smaller variance.
> 	* **higher solution efficiency**：
> 		* As discussed in Q2, the CG algorithm efficiently solves large-scale linear systems numerically, avoiding matrix inversion and reducing time complexity.
>
> ---
>
> **【W2】: Domains with no experimental data**
>
> **[W2Response]:**
>
> - We acknowledge that there are some causal inference scenarios where no RCT data is available. However, it is regrettable that this paper assumes the domain of marketing optimization (where a small amount of RCT data can be obtained).
> - Meanwhile, we believe that Bi-DFCL has certain value in other scenarios beyond marketing optimization where RCT data can be obtained even when extremely scarce. We conducted experiments in **Appendix B.3 (Details of In-depth Analysis), Table 6**, which demonstrate that our method still has high application value when RCT data is extremely sparse (e.g., only 1%).
> - However, scenarios without any available RCT data are beyond the scope of this paper, but this will be our future research direction. We apologize for this and hope the reviewers will understand. We greatly appreciate the pointing out of this issue.
>
> ---
>
> **【W3】:  How much does the Bridge Network alone contribute vs. the full bi-level setup?**
>
> **[W3Response]:** Thanks for this question. We apologize for the misunderstanding caused.
>
> * First, we clarify that the bridge model and the BLO are integrated, that is, **the column of the bi-level optimization problem in the table 3 of ablation study represents the contribution of the bridge model.**
> * Secondly, due to space constraints, we have placed the description of ablation study in **Appendix B.2 Details of Ablation Studies.**
> * We apologize for the ambiguity here, and in subsequent versions, we will move the detailed description of the ablation study in Appendix B.2 to the main text to facilitate better understanding for readers.
>
> ---
>
> **We sincerely thank you for your feedback and will provide more clarifications and explanations in the revised version, and welcome any further technical advice or questions on this work and we will make our best to address your concerns.**

---

> > ### Comment · Reviewer_HrqG · 2025-08-03
> >
> > Thanks for the response. I keep my score.

---

> > > ### Author Response · Authors · 2025-08-03
> > > **Thanks again!**
> > >
> > > Thanks for your response. We hope our answers have addressed your concerns. We will provide more clarifications and explanations in the revised version, and welcome any further technical advice or questions on this work. Thanks again!

---

### Official Review · Reviewer_LRH2 · 2025-07-03

**Clarity:** 3
**Significance:** 3
**Originality:** 3
**Rating:** 5
**Confidence:** 4

**Summary:**

The paper proposes Bi-Level Decision-Focused Causal Learning (Bi-DFCL), a novel framework for marketing optimization that addresses two key challenges: (1) the misalignment between prediction accuracy and decision quality in traditional two-stage methods (TSM), and (2) the bias-variance dilemma arising from the use of OBS and RCT data. This formulation enables the unbiased OR estimator to correct learning directions from biased OBS, achieving optimal bias-variance tradeoff.

**Questions:**

I have mentioned some questions in the weakness part above. Below are some more:

 I am very curious about how the data quantity and quality influences the results.  Could you discuss the convergence properties of this approach, especially when dealing with noisy or high-variance RCT data? What could happen if the method handle extreme cases where RCT data is extremely scarce (e.g., <1% of the dataset)? Are there safeguards against overfitting?

**Ethical Concerns:**

["NO or VERY MINOR ethics concerns only"]

**Final Justification:**

The authors carefully address my concerns on theoretical and computational results. What I really appreciate is that the authors honestly acknowledged their tiny weakness.

**Limitations:**

See above.

**Paper Formatting Concerns:**

N/A.

**Quality:**

3

**Strengths And Weaknesses:**

# Strength

1. I like the problem itself very much. The mismatch between estimating and decision-making, from my perspective, is one of the central questions in causal inference.
2. The bi-level optimization idea is insightful, together with the observation on the role of OBS and RCT data.The use of implicit differentiation for bi-level optimization is particularly interesting and novel.
3. The experimentation on multiple datasets validates the framework.

# Weakness
1. I am a bit worried about the theoretical ground for conducting inference. The framework itself is sophisticated, but the settings the authors care about requires very solid theoretical results for inference (e.g., central limit theorem results). I am not super clear how to do that Bi-DFCL.

2. While the paper addresses scalability, the bi-level optimization and implicit differentiation steps may still be computationally expensive for very large-scale applications. A more detailed discussion of runtime and resource requirements would be helpful.

3. The experiments focus on MTBAP, where we can easily have something like eq. (1) to define $\mathcal{L}_{DL}(\theta)$. Is it always possible to easily get the counterpart of eq.(1) for other problems?

---

> ### Author Rebuttal · Authors · 2025-07-30
>
> We sincerely appreciate your approval of the idea and the novelty of this work, and thank you for the helpful suggestions. Below, we hope to address your concerns and questions to improve the clarity and readability of our paper.
>
> ---
>
> ### **【W1】The theoretical ground of Bi-DFCL**
>
> **[W1]: ”I am a bit worried about the theoretical ground for conducting inference. The framework itself is sophisticated, but the settings the authors care about requires very solid theoretical results for inference. I am not super clear how to do that Bi-DFCL.”**
>
> **[W1Response]:** We will explain the framework of Bi-DFCL and its theoretical foundations from two perspectives: offline training and online inference.
>
> 1. **Offline Training**. To address the bias-variance dilemma and prediction-decision misalignment, we formulate a bi-level optimization problem using low-variance OBS data and unbiased RCT data: Find the optimal bridge network that debiases OBS data to enable the target network (lower-level) to minimize decision loss on RCT data (upper-level). This formulation is naturally and straightforwardly defined, requiring no additional theoretical assumptions. The primary technical challenges arise in gradient computation for the bridge network during upper-level optimization $\nabla\_{\phi}\mathcal{L}\_{\text {DL }}\left(\theta^{\star}(\phi) ; \mathcal{D}\_{\mathrm{RCT}}\right)$, specifically:
> 	1. **Gradient of Decision Loss**. As decision-making involves discrete optimization, the decision loss is non-differentiable. In Section 4.2, we resolve this via: a) Continuous relaxation using softmax functions. b) Black-box perturbation methods for gradient estimation.
> 	2. **Gradient of Target Network $\theta^*(\phi)$ w.r.t. Bridge Network $\phi$.** No closed-form solution exists for the target network $\theta^*(\phi)$, making direct gradient computation infeasible. In Section 4.3, We address this by: a) Implicit differentiation for gradient derivation. b) Conjugate gradient methods to ensure computational efficiency.
>
> 2. **Online Inference**. After offline training, the target network—whose architecture and parameters are identical to conventional OBS data models—is deployed directly for online inference.
>
> ---
>
> ### **【W2】The computational overhead**
>
> **[W2]: ”While the paper addresses scalability, the bi-level optimization and implicit differentiation steps may still be computationally expensive for very large-scale applications. A more detailed discussion of runtime and resource requirements would be helpful.”**
>
> **[W2Response]:** Thanks for this helpful advice.
>
> * **Space Efficiency**: Details of space efficiency and resource requirements can be find in **Appendix B.1.3 Experimental Details**, Our proposed Bi-DFCL does not incur additional space overhead compared to existing baselines.
>
> *  **Time Efficiency**:
>
> 	* **Online Inference:**  We only use the well-trained Target Model (see the Output of Algorithm 1) for online inference. Therefore, the inference time of our method is exactly the same as that of simple causal learning methods—because their model structures are the same.
>
> 	* **Offline Training:**
>
> 		* However, Bi-DFCL does introduce additional time overhead during offline training—which is one of the issues we have been working to address.
>
> 		1. **Experiments**
>
> 			  * As suggested by the reviewer, we compare the training time (minutes) of different baselines on Marketing data II.
>
> 			* | Dataset II |        Methods         | Training time |
> 				| :--------: | :--------------------: | :-----------: |
> 				|  RCT+OBS   |         TSM-SL         |    42.332     |
> 				|  RCT+OBS   |        KD-Label        |    67.358     |
> 				|  RCT+OBS   |         LTD-DR         |    492.559    |
> 				|  RCT+OBS   |       AutoDebias       |    397.886    |
> 				|  RCT+OBS   |    **Bi-DFCL-PPL**     |  **265.263**  |
> 				|  RCT+OBS   |    **Bi-DFCL-PIFD**    |  **294.927**  |
> 				|  RCT+OBS   | Bi-DFCL-PPL-wo ID Alg  |    345.132    |
> 				|  RCT+OBS   | Bi-DFCL-PIFD-wo ID Alg |    427.515    |
>
> 			  * Note: For fairness, all methods were fully trained for 500 epochs using the same model structure (no early stopping).
>
> 			* "Bi-DFCL-wo ID Alg" indicates that the solution to BLO problem no longer uses the implicit differentiation algorithm, but instead uses the more common explicit differentiation algorithm (see Appendix A.4 for details).
>
> 		2. **Theoretical Analysis**
>
> 			  * We decompose the additional time overhead of Bi-DFCL compared to TSM-SL into two parts: (1)the Decision Loss; (2)solving the BLO;
>
> 			  * Note that most of the time overhead of Bi-DFCL comes from solving the BLO.
>
> 			  * To accelerate this process, we use implicit differentiation and the CG algorithm. The latter efficiently solves large-scale linear systems numerically, avoiding matrix inversion and reducing time complexity from O(n³) to O(n).
>
> 		3. **Summary**
>
> 			  * Bi-DFCL does not incur additional space overhead or inference time overhead, but it does have a training time overhead 6-7 times that of the baseline TSM-SL.
>
> 			  * Bi-DFCL achieve a linear increase in training time with the number of samples.
>
> 			  * However, on the one hand, the extra offline training time is absolutely worthwhile in exchange for higher online decision quality. On the other hand, we have also made several improvements to address this issue (such as the CG algorithm).
>
> ---
>
> ### **【W3】Decision Loss for other problems**
>
> **[W3]: ”The experiments focus on MTBAP, where we can easily have something like eq. (1) to define decision loss. Is it always possible to easily get the counterpart of eq.(1) for other problems?”**
>
> **[W3Response]:** We explain the question through two optimization perspectives: common resource allocation problems and more complex formulations.
>
> 1. **For marketing resource allocation**:  typically formalized as knapsack problems (e.g., 0-1, multi-dimensional, or multiple-choice) — Lagrangian duality algorithms yield approximate solutions and decision loss that exhibit structural similarity to Eq.(1).
>
> 2. **For more complex optimization problems** : deriving closed-form solutions like Eq.(1) is not always feasible. Crucially, this limitation does not preclude gradient computation — our black-box perturbation method (PIFD) computes the gradients of decision loss without requiring closed-form solutions.
>
> ---
>
> ### **【Questions】**
>
> **【Q1】: ” Could you discuss the convergence properties of this approach, especially when dealing with noisy or high-variance RCT data? What could happen if the method handle extreme cases where RCT data is extremely scarce (e.g., <1% of the dataset)? Are there safeguards against overfitting?”**
>
> **[Q1Response]:** Good questions.
>
> 1. **The convergence properties where RCT data is extremely scarce**:
>
> 	* **1%~10%**
> 		* We have discussed in the main text and **Appendix B.3 (Details of In-depth Analysis)** the performance changes when the ratio of RCT to OBS data ranges from (1:10 to 1:100).
>
> 		* It can be seen that as the absolute amount of RCT data decreases, the performance of Bi-DFCL decreases to a certain extent, but it still outperforms most baseline methods.
> 	* **<1%**
> 		* For RCT data <1%, the key is its **absolute volume** rather than proportion, with two scenarios:
> 			1. OBS data expands but RCT data stays remains at hundreds of of thousands (e.g., 220,000)：Even though the proportion of RCT data <1%, Bi-DFCL’s performance does not deteriorate. Instead, it becomes more robust due to the increased OBS data.
> 			2. The absolute volume of RCT data continues to decrease (e.g.,  < 220,000) :
> 				* In experiments, we empirically set the batch size for decision loss at the tens-of-thousands level (e.g., 65,536). Thus, if there are still tens of thousands of RCT data (e.g., more than 0.1%), our method may still work (as discussed in Appendix B.3), because the decision loss constructed from tens of thousands of RCT data remains reasonably representative.
> 				* If the volume of RCT data is only a few thousand or less (less than 0.1%), it is inherently difficult to train a good model with such a small amount of RCT data, but we have also implemented safeguards against overfitting to prevent extremely poor performance (will be discussed in the following text).
> 2. **Are there safeguards against overfitting?**
> 	* Yes, this is an insightful question. We have indeed done a lot of work to combat high variance when RCT data is scarce, and the specific measures are as follows：
> 		1. **The counterfactual pseudo-labels of parameterized prediction loss.**
>
> 			* Counterfactual pseudo-labels are a key method to reduce training variance.
> 			* Unlike unstable existing reweighting methods (prone to extreme weights causing high variance), our parameterized prediction loss with such pseudo-labels is more stable.
> 		2. **Gating coefficients for stability and ensuring a baseline.**
> 			* Via knowledge distillation, the bridge model outputs gating coefficients to adaptively weight two teacher models’ outputs for counterfactual pseudo-labels.
> 			* This gating-based integration enhances stability, improves fault tolerance, mitigates high variance, and reduces warm-start needs.
> 		3. **The implicit differentiation algorithm.**
> 			* Its advantages (with the CG algorithm) include more accurate/stable gradients and higher efficiency.
> 			* It stabilizes training by avoiding explicit differentiation’s gradient vanishing, addressing high variance.
> 		4. **Other tricks**:
> 			* Setting the model structure of the bridge network to be relatively simple (e.g., reducing the number of layers) can effectively alleviate the overfitting phenomenon.
> 			* Updating the bridge network every k steps can also mitigate high variance.
>
> ---
>
> **We sincerely thank you for your feedback and will provide more clarifications and explanations in the revised version, and welcome any further technical advice or questions on this work and we will make our best to address your concerns.**

---

> > ### Comment · Reviewer_LRH2 · 2025-08-01
> > **Thank you for your time!**
> >
> > I want to sincerely thank the authors for their effort and time which efficiently resolve my concern. I will increase my score to 5. Thank you!

---

> > > ### Author Response · Authors · 2025-08-01
> > > **Thanks again!**
> > >
> > > We are glad to know that your concerns have been effectively addressed. We are very grateful for your constructive comments and questions, which helped improve the clarity and quality of our paper. Thanks again!

---

### Official Review · Reviewer_9osN · 2025-07-16

**Clarity:** 3
**Significance:** 3
**Originality:** 2
**Rating:** 5
**Confidence:** 3

**Summary:**

This paper proposes a new method, Bi-level Decision-Focused Causal Learning (Bi-DFCL), for effective marketing resource allocation. The proposed method leverages both observational and experimental data to address challenges in resource allocation problems, i.e., the prediction-decision misalignment and bias-variance dilemma.

**Questions:**

1)	I am wondering if the proposed method has been tested on appropriate datasets. The strength of randomized controlled trials (RCTs) lies in random assignment, which ensures that treatment is independent of potential outcomes. The proposed method also relies on the ignorability assumption of the experimental data to derive unbiased estimators. Here, one of my concerns is whether the training, validation, and test sets—derived from each of the datasets (CRITEO-UPLIFT v2, Marketing Data I, and Marketing Data II)—still uphold this assumption. Since the original datasets, which presumably satisfied the ignorability assumption, were divided arbitrarily into three subsets, I am concerned whether the resulting subsamples still retain the characteristics of RCTs. Are their control and treatment groups still balanced after the arbitrary division?

2)	Related to the above point, I am also wondering how the real-world A/B tests were designed. Since the proposed method relies on experimental data to derive unbiased estimates, ensuring that the experimental data meet RCT assumptions is critical; Otherwise, the performance results of the proposed method are tenuous. The authors conducted two experiments on the same food delivery platforms; One is about the money-off promotion, and the other is about discounts. Were they conducted independently and during different time periods? Were there any external shocks during the experiment (e.g., holidays, weather) that could have confounded results? Are the 180 and 192 features in each experiment pre-treatment covariates? In real deployments, various factors, other than treatment, can be involved and dramatically bias results. The authors may want to provide further details on the description of the real-world A/B tests.

3)	I would like to better understand how the proposed method stands up against the existing alternatives. In Table 2, several existing methods (e.g., AutoDebias, LTD-DR) also use both RCT and observational data. The authors may need to clarify how Bi-DFCL compares against these hybrid approaches (not only the ones that rely on either RCT or observational data, as the authors did) and what the pros and cons of these alternative approaches are.

**Ethical Concerns:**

["NO or VERY MINOR ethics concerns only"]

**Final Justification:**

My comments are mostly about missing, but essential information in the paper. Since the authors clarified all the questions I raised in the rebuttal, I keep my ratings.

**Limitations:**

yes

**Paper Formatting Concerns:**

Not observed.

**Quality:**

4

**Strengths And Weaknesses:**

The idea of utilizing the advantages of different types of data, i.e., experimental data for deriving unbiased estimators and observational data for acquiring more samples and generalizability, is brilliant, and the evaluation setup to verify the proposed method's effectiveness is commendable. However, a few clarifications and improvements are still needed, which are described in the section below.

---

> ### Author Rebuttal · Authors · 2025-07-30
>
> We sincerely appreciate your approval of the idea and the evaluation setup of this work, and thank you for the helpful suggestions. Below, we hope to address your concerns and questions to improve the clarity and readability of our paper.
>
> ---
>
>
>
> **[Q1]: “I am wondering if the proposed method has been tested on appropriate datasets. The strength of randomized controlled trials (RCTs) lies in random assignment, which ensures that treatment is independent of potential outcomes. The proposed method also relies on the ignorability assumption of the experimental data to derive unbiased estimators. Here, one of my concerns is whether the training, validation, and test sets—derived from each of the datasets (CRITEO-UPLIFT v2, Marketing Data I, and Marketing Data II)—still uphold this assumption. Since the original datasets, which presumably satisfied the ignorability assumption, were divided arbitrarily into three subsets, I am concerned whether the resulting subsamples still retain the characteristics of RCTs. Are their control and treatment groups still balanced after the arbitrary division?”**
>
> **[Q1Response]:**  We confirm that all dataset splits maintain RCT properties through three evidence-based guarantees:
>
> 1. **Original RCT Collection Mechanism**: Treatments were assigned purely randomly independent of pre-treatment covariates $X$, ensuring: $(Y_i(0),Y_i(1)) \perp T$ and $X\perp T$.
>
> 2. **Stratified Random Splitting Procedure**: For each dataset (CRITEO-UPLIFT v2, Marketing Data I/II), we performed covariate-agnostic random splitting: Train/validation/test partitions created via simple random sampling. Splitting criteria excluded all $X$ variables, thus still preserving: $(Y_i(0),Y_i(1)) \perp T$ and $X\perp T$.
>
> 3. **Post-Splitting Balance Verification**: Prior to model training and testing, we assessed covariate balance across treatment groups using Standardized Mean Difference (SMD). Across all datasets, SMD values were consistently below 0.1 (standard balance threshold), confirming balanced covariate distributions between treatment groups.
>
> ---
>
>
>
> **[Q2] : “Related to the above point, I am also wondering how the real-world A/B tests were designed. Since the proposed method relies on experimental data to derive unbiased estimates, ensuring that the experimental data meet RCT assumptions is critical; Otherwise, the performance results of the proposed method are tenuous. The authors conducted two experiments on the same food delivery platforms; One is about the money-off promotion, and the other is about discounts. Were they conducted independently and during different time periods? Were there any external shocks during the experiment (e.g., holidays, weather) that could have confounded results? Are the 180 and 192 features in each experiment pre-treatment covariates? In real deployments, various factors, other than treatment, can be involved and dramatically bias results. The authors may want to provide further details on the description of the real-world A/B tests.”**
>
> **[Q2Response]:** Additional Details on Offline Marketing Datasets and Online A/B Experiment:
>
> On the online food delivery platform, we collected two offline marketing campaign datasets and conducted one online A/B test. These datasets and the online experiment were conducted independently at different times.
>
> 1. **Marketing Data I**: Collected during August–September 2024, corresponding to a money-off campaign.
>
> 2. **Marketing Data II**: Collected in March 2025, corresponding to a discount campaign. The campaign format difference stems from product strategy adjustments implemented in January 2025.
>
> 3. **The online A/B experiment** was conducted in April 2025 using the model trained on **Marketing Data II** and aligned with the discount campaign.
>
> For both RCT data (from Marketing Data I-II) and the online A/B experiment, some external factors (e.g., holidays, weather, supply-demand fluctuations, or competitor actions) may have existed during the experimental periods. However, the treatment assignments (in offline RCT data) and experiment groups (in online A/B tests) were independent of all covariates—both observed and unobserved. Thus, external factors affected all treatment/experiment groups equally and the balance assumptions for RCTs and A/B tests remain valid.
>
> All features used in the offline marketing datasets and online A/B tests (180 features for Marketing Data I; 192 for Marketing Data II and online tests) are pre-treatment covariates.
>
> ---
>
>
>
> **[Q3] :“I would like to better understand how the proposed method stands up against the existing alternatives. In Table 2, several existing methods (e.g., AutoDebias, LTD-DR) also use both RCT and observational data. The authors may need to clarify how Bi-DFCL compares against these hybrid approaches (not only the ones that rely on either RCT or observational data, as the authors did) and what the pros and cons of these alternative approaches are.”**
>
> **[Q3Response]:** Thanks for the question.
>
> * It is true that some existing methods (e.g.,CausE, KD, AutoDebias, LTD-DR) also propose to utilize RCT and OBS data more effectively, which shares similarities with our idea.
> * However, our method has the following innovations compared to these approaches:
> 	1. **Innovation in addressing prediction-decision misalignment**:
> 		* First, one of the biggest differences between our method and these baselines that also utilize RCT+OBS data is that we consider the issue of prediction-decision misalignment.
> 		* As stated in the main text, prediction-decision misalignment is a challenging problem in marketing optimization scenarios. However, **existing RCT+OBS baselines only focus on improving prediction accuracy**, which will encounter severe prediction-decision misalignment problems in actual marketing optimization and causal inference scenarios.
> 		* We have solved this by deriving two innovative differentiable decision losses— PPL and PIFD, and we have further innovated in addressing the prediction-decision misalignment itself by designing decision losses based on the primal problem.
> 	2. **Innovation in the form of the parameterized prediction loss**:
> 		* Second, we have also made innovations in the construction of the parameterized prediction loss.
> 		* **Most existing methods rely on reweighting the loss**, such as IPS, AutoDebias, LTD.
> 		* In practice, we found that this reweighting method is very unstable in effect — because the weights may be learned to be close to 0 or very large — thus causing excessive changes to the loss and resulting in high variance and instability.
> 		* To address this issue, we use a parameterized prediction loss with **counterfactual pseudo-labels**, which are more stable.
> 		* In addition, we also incorporate knowledge distillation — instead of directly outputting pseudo-labels, we enable the bridge model to output **gating coefficients**, thereby adaptively weighting the outputs of two different teacher models to combine and obtain counterfactual pseudo-labels.
> 		* The pseudo-labels generated by this gating-based method are more stable, mitigating the problem of high variance and the need for warm starts.
> 	3. **Innovation in solving BLO problems—Implicit Differentiation+CG Algorithm**:
> 		* Third, we have innovated in solving bi-level optimization problems.
> 		* Existing methods mainly use explicit differentiation, which assumes optimality can be achieved in one gradient descent step (see Appendix A.4) but is path-dependent, prone to vanishing gradients and suboptimal solutions—unrealistic in practice, as single updates yield suboptimal results and multiple updates exacerbate gradient vanishing.
> 		* We instead adopt implicit differentiation, which is path-independent, yielding more accurate and stable gradients.
> 		* While it faces time/space overhead, we address this with the CG algorithm, avoiding large matrix storage and reducing time complexity from O(n³) to O(n).
> 		* In summary, **our algorithm for solving bi-level optimization problems can achieve more accurate and stable solutions with higher efficiency**.
> 	4. **Innovation in real-world application scenarios**：
> 		* Finally, it is worth emphasizing that existing RCT+OBS baseline studies (e.g., CausE, KD, AutoDebias, LTD-DR) focus on public datasets for recommendation systems, with no methods truly applicable to large-scale industrial scenarios of marketing optimization and causal inference.
> 		* In contrast, our method targets two key challenges in these areas, with its effectiveness validated by multiple offline experiments and online A/B tests. It has been deployed in real scenarios, generating business benefits.
>
> ---
>
> **We sincerely thank you for your feedback and will provide more clarifications and explanations in the revised version, and welcome any further technical advice or questions on this work and we will make our best to address your concerns.**

---

> > ### Comment · Reviewer_9osN · 2025-08-04
> >
> > I want to sincerely thank the authors for answering all of my questions. My concerns are all well addressed and resolved. Thank you!

---

> > > ### Author Response · Authors · 2025-08-05
> > > **Thanks again!**
> > >
> > > We are glad to know that all your concerns have been effectively addressed. We are very grateful for your constructive comments and questions, which helped improve the clarity and quality of our paper. Thanks again!

---

### Official Review · Reviewer_pm1M · 2025-07-17

**Clarity:** 2
**Significance:** 3
**Originality:** 2
**Rating:** 3
**Confidence:** 4

**Summary:**

The paper proposes a Bi-Level Decision-Focused Causal Learning (Bi-DFCL) framework for large-scale marketing optimization. It addresses two key challenges in existing approaches: prediction decision misalignment and bias-variance dilemma. The paper assumes availability of both RCT and observational data and combines tasks of: a) prediction and b) decision to recommend optimal allocation under constraints. The paper performed extensive experiments on a diverse set of data sources to support their claims.

**Questions:**

The main questions I had were:
1. Ln180: why would L_{PL} benefit from training only on observational data? Wouldn't the prediction loss also benefit from being trained on RCT data (if available)? That way the quality (in terms of accuracy) of counterfactual allocations can be measured correctly.
2. Given RCT data + OBS data, is it not possible to train an ITE estimator (possibly for multi-treatment) and use it for L_{DL}? Is there any advantage of the proposed approach over this simplistic approach?
3. The paper does not discuss the amount of RCT data available vs amount of OBS data. A practical assumption would be that RCT data is much smaller in size as compared to observational data. Also, the experimental setup (Table1) seems to have similar orders of RCT and observational training data. Is there any theoretical or experimental results which shows the performance where the RCT data is orders of magnitude smaller than the observational data (which is the expected setup in practice)?

**Ethical Concerns:**

["NO or VERY MINOR ethics concerns only"]

**Limitations:**

Yes

**Quality:**

2

**Strengths And Weaknesses:**

Strengths:
1. The paper discusses an important problem for marketing optimisations.
2. The paper is well written and the approach is clearly presented.

Weaknesses:
1. The paper does not discuss why the prediction loss should be trained only on observational data. In the presence of observational + RCT data, it is much more efficient to train ITE estimators and use them for downstream optimisation.
2. The experimental setup (Table1) uses a similar amount of RCT and observational data which seems impractical in most scenarios.
3. Claims in Section 4.1 does not seem to be correct. If there is ample amount of RCT data available, then both L_{PL} and L_{DL} would benefit from it i.e., RCT data does not selectively improve only one of these losses.

---

> ### Author Rebuttal · Authors · 2025-07-26
>
> We sincerely appreciate the reviewer’s great efforts and comments.  Below, we hope to address your concerns and questions to improve the clarity and readability of our paper.
>
> ---
>
>
>
> ### **【Q3】The amount of RCT data vs OBS data**
>
> **[Q3.1]: “The paper does not discuss the amount of RCT data vs OBS data”**
>
> **[Q3.2]: “The Table1 seems to have similar orders of RCT and OBS training data”**
>
> **[Q3.3]: “The performance where the RCT data is orders of magnitude smaller than OBS data”**
>
> **[Q3Response]**: We apologize for the misunderstanding caused.
>
> 1. **We have discussed the RCT-OBS ratios from 1:10 to 1:100.**
> 	* We clarify that the amount between RCT and OBS data **have been discussed** in both the main text and Appendix, with various ratios (**1:10–>1:100**).
>
> 	* In **Table 1 of the main text** , we present three datasets with distinct RCT:OBS training data ratios :
> 		* CRITEO: **~1:5** (698,980 RCT vs. 3,498,294 OBS)
> 		* Marketing Data I: **~1:10** (2,220,781 RCT vs. 22,201,405 OBS)
> 		* Marketing Data II: **~1:17** (2,017,450 RCT vs. 33,815,274 OBS)
>
> 	* In **Supplementary Material: Appendix B.3 (Details of In-depth Analysis), Table 6** supplements these results with additional ratios:
> 		* Marketing Data I: **~1:20** (1,100,000 RCT vs. 22,201,405 OBS)
> 		* Marketing Data I: **~1:100** (222,000 RCT vs. 22,201,405 OBS)
>
> 	* These all validate Bi-DFCL’s performance under extreme RCT scarcity (ratios of 1:10–>1:100) .
>
> 2. **The 1:10~1:100 of RCT-OBS ratios in the real marketing scenarios are common and representative**.
> 	* We believe that the statement that "the RCT and OBS data in Table 1 are of similar orders" may be a **misunderstanding**.
>
> 	* **1:10~1:100 of RCT-OBS ratios in our experiments are representative of real marketing scenarios** , which are consistent with standard practices in the large-scale marketing optimization— **RCT data is typically collected from 1~ 10% of platform traffic**. **This results that RCT are 10–100 times smaller than OBS data**.
>
> 3. **Evidence: Amount of our RCT data are also consistent with existing methods**.
> 	* Existing SOTA methods for large-scale marketing optimization (e.g. DRP[1], RERUM[2],  KUAI-NES[3], DHCL [4], DFCL [5]) typically rely on **millions of RCT data** for training.
> 	* **Our RCT data volumes (2-3million RCT samples) are consistent with these precedents**.
>
> ---
>
>
>
> ### **【Q1】$L\_{PL}$  trained only on OBS data or OBS+RCT data**
>
> **[Q1]: “Why would $L\_{PL}$ benefit from only on obs data? Wouldn't $L\_{PL}$ also benefit from being trained on RCT data ? ”**
>
> **[Q1Response]：**
>
> 1. **Practical Evidence:  small gains of directly adding RCT data to  $L\_{PL}$**.
>
> 	*  While $L\_{PL}$ can use a mix of OBS+RCT data, the incremental benefit of adding RCT data is almost negligible in practice due to the sheer scale of OBS data.
>
> 	*  As shown in Table 2 , TSM-SL exhibit minimal improvements when trained on OBS+RCT compared to OBS-only:
>
> 		-  Marketing Data I: TSM-SL (OBS) achieves an EOM of 1.0067, while TSM-SL (OBS+RCT) improves to 1.0071—a little gain of **0.04%**.
> 		-  Marketing Data II: TSM-SL (OBS) achieves 0.9957, while TSM-SL (OBS+RCT) reaches 0.9988—a little gain of **0.31%**.
>
> 	*  These results align with our expectation: since OBS data is magnitude larger, RCT data contributes little.
>
> 2. **Theoretical Analysis: Do not conflict with our assumptions**
>
> 	* We agree that using mixed data to train $L\_{PL}$  may be a little helpful, but it does not conflict with Bi-DFCL.
>
> 	* **Our key assumption is that $L\_{DL}$ must use RCT data,  $L\_{PL}$ can use any types of data.   We did not state that $L\_{PL}$ *cannot* use RCT data**.
>
> 	* Because the core difference between RCT and OBS lies in whether X and T are orthogonal. If RCT and OBS are simply mixed, X and T remain non-orthogonal. Therefore, **from a theoretical perspective, RCT + OBS is equivalent to OBS**.
> 		* OBS data is inherently biased due to non-random treatment assignment ( X $\not\perp$ T \), while RCT data is unbiased because of randomization (X $ \perp$ T\)
> 		* However, when combined, the overwhelming volume of OBS data (10–100x RCT data) ensures the mixed dataset retains X $ \not\perp$ T overall. Thus, mixed data does not resolve the fundamental bias in OBS data
>
> ---
>
>
>
> ### **【Q2&&W3】Given RCT+OBS, Train an ITE estimator using $L\_{DL}$?**
>
> **[Q2.1]: “Given RCT+OBS data, is it not possible to train an ITE estimator using $L\_{DL}$ ? ”**
>
> **[Q2.2]: “Is there any advantage over this simplistic approach ? ”**
>
> **[W3] : “If there is ample amount of RCT data, then both $L\_{PL}$  and $L\_{DL}$ would benefit from it , RCT data does not selectively improve only one. ”**
>
> **[Q2&&W3Response]**：Good question.
>
> 1. **[Q2.1Response]: $L\_{DL}$ must be trained exclusively on RCT data**.
>
> 	1. **Theoretical Analysis**:
>
> 		* **Unbiased OR Estimator**:
>
> 			* $L\_{DL}$ serves as an unbiased estimator of decision quality, and its derivation critically relies on the unbiased nature of RCT data (as proven in Appendix A.2).
>
> 			* **If using biased OBS data or OBS+RCT data ( X $ \not\perp$ T ) , $L\_{DL}$  is absolutely a wrong  and useless loss**.
>
> 		* **Extremely Sensitive To Bias**:
>
> 			* As discussed in Section 4.1—“$L\_{DL}$ is highly dependent on the unbiasedness of RCT: minimizing $L\_{DL}$ on biased OBS would greatly amplify bias and severely degrade decision quality”.
> 			* OBS+RCT data still suffers from huge bias, which is almost equivalent to only OBS.
>
> 	2. **Practical Experiment**:
>
> 		* Note that when using OBS or OBS+RCT for $L\_{DL}$  , the performance becomes extremely poor.
>
> 		* |  Data   |      Loss      | EOM (dataset I) | EOM (dataset II) |
> 			| :-----: | :------------: | :-------------: | :--------------: |
> 			|   RCT   |   $L\_{PL}$    |     1.0000      |      1.0000      |
> 			|   RCT   | $L\_{DL-PIFD}$ |     1.0170      |      1.0153      |
> 			|   OBS   |   $L\_{PL}$    |     1.0067      |      0.9957      |
> 			|   OBS   | $L\_{DL-PIFD}$ |   **0.9703**    |    **0.9542**    |
> 			| RCT+OBS |   $L\_{PL}$    |     1.0071      |      0.9988      |
> 			| RCT+OBS | $L\_{DL-PIFD}$ |   **0.9727**    |    **0.9561**    |
>
> 	3. **Evidence: Custom loss always use only RCT from existing works. (see Section 2)**
>
> 		* Existing studies that address prediction-decision misalignment by **constructing custom losses (e.g., DRP[1], RERUM[2],  KUAI-NES[3], DHCL [4], DFCL [5]) exclusively rely on RCT data**. In their frameworks, no OBS data is leveraged, as these custom losses—similar to our $L\_{DL}$ —**are inherently dependent on the unbiasedness of RCT to ensure valid decision quality estimation**.
>
> 		* In contrast, **methods that do utilize OBS data** (e.g., IPS [6], DragonNet [7]) **are restricted to standard loss (e.g., MSE) for prediction**, which leads to prediction-decision misalignment.
>
> 2. **[Q2.2Response]: Advantage of Bi-DFCL over this simplistic approach**.
>
> 	*  As discussed above, **simplistic approach does not work—$L\_{DL}$ must be trained exclusively on RCT data**. Existing works either use RCT+custom loss or use OBS+standard loss.
>
> 	*  **One key contribution of Bi-DFCL is solving this bias-variance tradeoff**: it effectively incorporates OBS data to reduce variance  (via $L\_{PL}$ in the lower level) while using $L\_{DL}$  (must trained on RCT in the upper level) to calibrate biases of OBS data. **This design retains the unbiased decision estimation of  $L\_{DL}$  while harnessing OBS data for improved generalization—addressing both limitations of existing works**.
>
> 3. **[W3Response]: The extent to which RCT data benefits $L\_{PL}$ and $L\_{DL}$  differs significantly**.
>
> 	* Yes. We agree that both  $L\_{PL}$  and  $L\_{DL}$ can benefit from RCT data when it is abundant.
>
> 	* However, **there is a significant difference in the degree of their dependence on RCT data**:
>
> 		1. **RCT is crucial for  $L\_{DL}$ ; without RCT data, $L\_{DL}$ becomes unusable**.
>
> 			* **Experiments:**
> 				* Marketing data I: 0.9703(OBS+$L\_{DL}$) —> 0.9727(OBS+RCT+$L\_{DL}$) —> 1.0170(RCT+$L\_{DL}$)
> 				* Marketing data II: 0.9542(OBS+$L\_{DL}$) —> 0.9561(OBS+RCT+$L\_{DL}$) —> 1.0156(RCT+$L\_{DL}$)
>
> 			* **Explaination**:   $L\_{DL}$ is extremely sensitive to bias of OBS data.
> 				* **$L\_{DL}$ optimizes the overall optimization problem** rather than the prediction error of distance at the individual sample level. Therefore, relative relationships between different samples interfere with each other.
> 				* Biased data will cause $L\_{DL}$ to severely distort the directions of the entire optimization problem, **$L\_{DL}$ becomes a wrong loss when using OBS or OBS+RCT.**
> 				* **So unbiased RCT data(without any OBS data) helps $L\_{DL}$ very much.**
>
> 		2. In contrast, **the benefit of RCT data to  $L\_{PL}$  is rather limited and can be negligible.** (Of course, we also support your approach of training  $L\_{PL}$  using a mixture of RCT+OBS data).
>
> 			* **Experiments:**
> 				* Marketing data I: 1.0067(OBS+$L\_{PL}$ ) —> 1.0071(OBS&RCT+$L\_{PL}$ ) —> 1.0000(RCT+$L\_{PL}$ )
> 				* Marketing data II: 0.9957(OBS+$L\_{PL}$ ) —> 0.9988(OBS&RCT+$L\_{PL}$ ) —> 1.0000(RCT+$L\_{PL}$ )
>
> 			* **We can see that 1/10～1/20 RCT data helps $L\_{PL}$ relatively little.**
>
> ---
>
> **We hope the above discussion will fully address your concerns, and we would really appreciate it if you could be generous in raising your score.** We look forward to your insightful and constructive responses to further help us. Thank you!
>
> ---
>
> ### **References**
>
> [1] Improve User Retention with Causal Learning
>
> [2] Rankability-enhanced revenue uplift modeling framework for online marketing
>
> [3] An End-to-End Framework for Marketing Effectiveness Optimization
>
> [4] Direct heterogeneous causal learning for resource allocation problems in marketing
>
> [5] Decision focused causal learning for direct counterfactual marketing optimization
>
> [6] Recommendations as treatments: Debiasing learning and evaluation
>
> [7] Adapting neural networks for the estimation of treatment effects

---

> > ### Comment · Reviewer_pm1M · 2025-08-04
> > **Thank you for your time**
> >
> > I would sincerely thank the authors for a detailed reply to my questions.
> >
> > I would keep my original score primarily since:
> > a) the RCT data is not used in the correct way in the response to Q1,
> > b) the response to Q2 does not address the primary concern - why can't an ITE estimator (trained on observational data) be used for $L_{DL}$?

---

> > > ### Author Response · Authors · 2025-08-08
> > >
> > > **Dear Reviewer pm1M**,
> > >
> > > We hope this message finds you well. As the discussion period is nearing its end with **one day remaining**, We wanted to ensure we have addressed all your concerns satisfactorily.
> > >
> > > **Could you please check whether our recent Clarification properly addressed your concerns?** We believe we have not only clearly articulated our responses to the two remaining issues (a and b) in this clarification, but also **explained your doubts from multiple perspectives**. We will be online waiting for your feedback.
> > >
> > > **If there are any additional points or feedback you'd like us to consider, please let us know.** Your insights are invaluable to us, and we're eager to address any remaining issues to improve our work.
> > >
> > > Thank you for your time and effort in reviewing our paper.
> > >
> > > The Authors of Paper 10859.

---

> ### Author Response · Authors · 2025-08-04
> **Clarification of Reviewer pm1M’s Concerns (a and b)**
>
> Thank you for your time to review our paper and detailed feedback. We believe there are some misunderstandings, which we would like to clarify below.
>
> ---
>
>
>
> ### **【a】The RCT data is not used in the correct way**
>
> **[Response]：** **We believe this is a misunderstanding—we are confident that we have correctly utilized RCT data to optimize decision quality (not prediction accuracy).**
>
> * We have emphasized in the rebuttal that any data can be used for $L\_{PL}$ , and even any novel models can be adopted for $L\_{PL}$ —this is not in conflict with our Bi-DFCL, as **higher prediction accuracy does not necessarily lead to better decision quality—which is precisely the motivation of our work**. Our experiments also provide evidence for this.
> * Due to prediction-decision misalignment, **our use of RCT data targets end-to-end enhancement of decision quality, not prediction accuracy**. In Bi-DFCL, **the lower-level $L\_{PL}$ is not standard MSE but a dynamically parameterized $L\_{PL}(\phi,\theta)$. Its purpose is not to boost prediction accuracy, but to improve generalization and therefore assist $L\_{DL}$ in jointly enhancing decision quality.**
> * We acknowledge that incorporating RCT data into OBS data can reduce MSE and improve prediction accuracy through various methods — **this may be what you refer to as the "correct way" to use RCT data, but such approaches are oriented toward improving prediction accuracy and do not necessarily translate to better decision quality.**
>
> ---
>
>
>
> ### **【b】why can't an OBS ITE estimator be used for $L\_{DL}$ ?**
>
> **[Response]：We believe this question has  been addressed in the rebuttal, and we provide additional details here.**
>
> 1. **Origin of $L\_{DL}$**:
>
> 	* **$L\_{DL}$ stems from academic research on decision-focused learning (DFL)[8],**  whose core idea is to optimize models end-to-end using $L\_{DL}$—a loss that reflects the real-time decision performance of predictive models in downstream tasks.
>
> 	* However, **$L\_{DL}$ requires labels across the entire sample space to compute the loss, making it inapplicable directly to causal inference problems[5]**—due to counterfactuals, we only have factual labels.
>
> 	* Fortunately, **some studies(e.g., DFCL[5] ,KUAI-NES[3]) have constructed unbiased estimates of $L\_{DL}$ by introducing EOM[9] using RCT data.**
>
> 	* In summary, applying $L\_{DL}$ to causal inference is non-trivial; while efforts exist to tackle this, currently, the only way to obtain unbiased estimates of $L\_{DL}$​​ for causal inference is to use RCT data.
>
> 2. **Consequences of using OBS data for $L\_{DL}$**:
>
> 	*  **$L\_{DL}$ is extremely sensitive to bias**—using biased OBS data would significantly amplify such bias, severely degrading decision quality.
>
> 	* Unlike standard losses that operate on individual samples, **$L\_{DL}$  treats the entire set of samples in each batch as a discrete optimization (OR) problem (e.g., MTBAP in this paper)**; its core goal is to maximize the unbiased estimation of decision quality derived from EOM, where EOM reflects the decision outcomes based on the current ITE estimator and OR algorithm (relying on large-scale RCT data for unbiased estimates of per-capita revenue or cost). **Using OBS data to calculate these metrics would invalidate EOM, fundamentally undermining $L\_{DL}$​‘ s objective.**
> 	* **More specifically, there is a striking difference in how $L\_{PL}$ and $L\_{DL}$ respond to bias**:
> 		* **$L\_{PL}$  operate on individual samples**：When bias exists in $L\_{PL}$, minimizing it still improves prediction accuracy (albeit accuracy for a biased data distribution rather than the target distribution).
> 		*  **$L\_{DL}$ operate on the entire OR problem**：When bias is present in $L\_{DL}$ , minimizing it no longer optimizes decision effectiveness at all—it may not correspond to decision performance under any real-world data distribution.
>
> 3. **Additional evidence**:
>
> 	* **Findings from existing literature (e.g., DRP[1], RERUM[2],  KUAI-NES[3], DHCL [4], DFCL [5]) confirm that constructing custom losses like $L\_{DL}$  all rely on RCT data.**
>
> 	* Applying $L\_{DL}$ to OBS data is a meaningful research topic, but it remains unsolved to date. This will also be the focus of our future work.
>
> ---
>
> We hope the above discussion will fully address your concerns. Thanks again for your time to review our paper and detailed feedback.  Welcome any further technical advice or questions on this work and we will make our best to address your concerns.
>
> ---
>
> ### **References**
>
> [8] Decision-focused learning: Foundations, state of the art, benchmark and future opportunities
>
> [9] Lbcf: A large-scale budget-constrained causal forest algorithm.

---

### Official Review · Reviewer_rQbD · 2025-07-19

**Clarity:** 3
**Significance:** 4
**Originality:** 4
**Rating:** 5
**Confidence:** 3

**Summary:**

The paper proposes Bi-Level Decision-Focused Causal Learning (Bi-DFCL), a framework for large-scale marketing optimization that addresses prediction-decision misalignment and bias-variance tradeoffs by leveraging both observational and experimental data. It introduces an unbiased estimator of decision quality and a bi-level optimization framework to dynamically correct learning directions, achieving superior performance in offline and online tests.

**Questions:**

- While Bi-DFCL demonstrates superior performance, could the authors elaborate on the computational overhead of the bi-level optimization compared to simpler causal methods? Real-time deployment constraints in marketing systems are critical.

- Can authors discuss further on how would it perform in domains with no experimental data (e.g., healthcare), given the paper’s focus on marketing?

**Ethical Concerns:**

["NO or VERY MINOR ethics concerns only"]

**Final Justification:**

This authors have solved my concerns in the time complexity and potential applications in scenarios where experimental data is less available. Thus I will keep my evaluation that this paper should be accepted.

**Limitations:**

yes

**Quality:**

4

**Strengths And Weaknesses:**

Strengths:

- The idea of addressing prediction-decision misalignment and bias-variance tradeoffs is novel.

- This method has been employed in real-world application and has demonstrates=d significant revenue improvements (e.g., 3.5% lift in orders), proving scalability and practicality.

Weakness:
 - The generalization scope could be a weakness. This method highly relies on hybrid RCT/OBS data, which might restrict applicability to platforms where experimental data is extremely scarce.

- Some key components of this paper, e.g., sensitivity of the hyper-parameters are expected to appear in the main text. However, authors put them into appendix.

---

> ### Author Rebuttal · Authors · 2025-07-27
>
> We sincerely appreciate your approval of the idea and the novelty of this work, and thank you for the helpful suggestions. Below, we hope to address your concerns and questions to improve the clarity and readability of our paper.
>
> ---
>
> ### **【Q1】The  computational overhead**
>
> **[Q1]: ”While Bi-DFCL demonstrates superior performance, could the authors elaborate on the computational overhead of the bi-level optimization compared to simpler causal methods? Real-time deployment constraints in marketing systems are critical.”**
>
> **[Q1Response]:** Thanks for the question.
>
> 1. **Space Efficiency**: Details of space efficiency can be find in **Appendix B.1.3 Experimental Details**, Our proposed Bi-DFCL **does not incur additional space overhead** compared to existing baselines.
>
> 	* Note that implicit differentiation algorithm itself requires the storage of large-scale inverse matrices, which have high demands on computing resources.
> 	* To enable the application of implicit differentiation in large-scale marketing optimization, we use the **Conjugate Gradient Algorithm (CG)** to avoid this problem—the conjugate gradient method **avoids the storage of large-scale inverse matrices through the trick of matrix-vector products**(see lines 239-245 in the main text and Appendix A.5 for specific details).
>
> 2. **Time Efficiency**:
>
> 	* **Online Inference:**
>
> 		* We first want to clarify that for the online deployment and inference of Bi-DFCL, we only use the well-trained Target Model (see the Output of Algorithm 1).
> 		* Therefore, **the inference time of our method is exactly the same as that of simple causal learning methods—because their model structures are the same**.
>
> 	* **Offline Training:**
>
> 		* However, Bi-DFCL does introduce additional time overhead during offline training—which is one of the issues we have been working to address. We analyze the time overhead of Bi-DFCL from both experimental and theoretical perspectives.
>
> 		1. **Experiments of Time Effciency**
>
> 			* As suggested by the reviewer, we compare the training time (minutes) of different baselines (using RCT + OBS data for training) on the Marketing data II. The results are shown below.
>
> 			* | Dataset II | Methods          | Training time |
> 				| :--------: | :--------------- | :-----------: |
> 				|  RCT+OBS   | TSM-SL           |    42.332     |
> 				|  RCT+OBS   | KD-Label         |    67.358     |
> 				|  RCT+OBS   | LTD-DR           |    492.559    |
> 				|  RCT+OBS   | AutoDebias       |    397.886    |
> 				|  RCT+OBS   | **Bi-DFCL-PPL**  |  **265.263**  |
> 				|  RCT+OBS   | **Bi-DFCL-PIFD** |  **294.927**  |
>
> 			* Note: For fairness, all methods were fully trained for 500 epochs using the same model structure (no early stopping).
>
> 			* From the table, the following conclusions can be drawn:
>
> 				* As one of the simplest causal learning method, TSM-SL has a training time of approximately 42 minutes.
> 				* The training time of Bi-DFCL is about **6～7 times that of TSM-SL**.
>
> 		2. **Theoretical Analysis about Time Effciency**
>
> 			* We decompose the additional time overhead of Bi-DFCL compared to TSM-SL into two parts:
>
> 				1. The time overhead of the Decision Loss;
> 				2. The time overhead of solving the bi-level optimization problem;
>
> 			*  We supplement the following ablation experiments:
>
> 			* | Dataset II |        Methods         | Training time |
> 				| :--------: | :--------------------: | :-----------: |
> 				|    RCT     |         TSM-SL         |     2.505     |
> 				|    RCT     |        DFCL-PPL        |     3.163     |
> 				|    RCT     |       DFCL-PIFD        |     9.948     |
> 				|    OBS     |         TSM-SL         |    39.918     |
> 				|  RCT+OBS   |         TSM-SL         |    42.332     |
> 				|  RCT+OBS   |    **Bi-DFCL-PPL**     |  **265.263**  |
> 				|  RCT+OBS   |    **Bi-DFCL-PIFD**    |  **294.927**  |
> 				|  RCT+OBS   | Bi-DFCL-PPL-wo ID Alg  |    345.132    |
> 				|  RCT+OBS   | Bi-DFCL-PIFD-wo ID Alg |    427.515    |
>
> 			* It can be seen that most of the time overhead of Bi-DFCL comes from solving the bi-level optimization problem.
>
> 			* To accelerate this process, we use the implicit differentiation algorithm and CG algorithm for solving, and it can be observed that **one of the advantages of implicit differentiation is lower time overhead**.
>
> 			* "**Bi-DFCL-wo ID Alg**" indicates that the solution to the bi-level optimization problem no longer uses the implicit differentiation algorithm, but instead uses the more common explicit differentiation algorithm (see Appendix A.4 for specific details).
>
> 			* **What makes the implicit differentiation algorithm faster?**:
>
> 				* The CG algorithm efficiently solves the following large-scale linear system through numerical methods, thereby avoiding matrix inversion — reducing the time complexity from O(n³) to O(n).
> 				* Implicit differentiation, as it does not depend on the optimization path, can obtain a more accurate optimal solution for the lower-level optimization problem compared to explicit differentiation(assuming one step GD to reach optimal). This allows solving the bi-level optimization problem once every k batches, rather than having k=1 as in explicit differentiation.
>
> 		3. **Summary**
>
> 			* Bi-DFCL does not incur additional space overhead or inference time overhead, but it **does have a training time overhead 6-7 times that of the baseline TSM-SL**.
> 			* However, on the one hand, the extra  **offline training time is absolutely worthwhile in exchange for higher online decision quality**. On the other hand, we have **also made several improvements to address this issue (such as the CG algorithm)**.
>
>
>
> ---
>
> ### **【Q2】Domains with no experimental data**
>
> **[Q2]: ”Can authors discuss further on how would it perform in domains with no experimental data (e.g., healthcare), given the paper’s focus on marketing?”**
>
> **[Q2Response]:**
>
> * We acknowledge that there are some causal inference scenarios where no RCT data is available. However, it is regrettable that this paper assumes the domain of marketing optimization (where a small amount of RCT data can be obtained).
> * Meanwhile, we believe that Bi-DFCL has certain value in other scenarios beyond marketing optimization where RCT data can be obtained even when extremely scarce. We conducted experiments in **Appendix B.3 (Details of In-depth Analysis), Table 6**, which demonstrate that our method still has high application value when RCT data is extremely sparse (e.g., only 1%).
> * However, scenarios without any available RCT data are beyond the scope of this paper, but this will be our future research direction. We apologize for this and hope the reviewers will understand. We greatly appreciate the pointing out of this issue.
>
> ---
>
> **【W2】: ”Some key components of this paper, e.g., sensitivity of the hyper-parameters are expected to appear in the main text. However, authors put them into appendix.”**
>
> **[W2Response]:** We appreciate the reviewer pointing this out. Due to page limitations during submission, some important content had to be placed in the appendix. In the subsequent revision, we will definitely move the important content from the appendix to the main text. Thank you again for the reviewer's suggestion.
>
> ---
>
> **We sincerely thank you for your feedback and will provide more clarifications and explanations in the revised version, and welcome any further technical advice or questions on this work and we will make our best to address your concerns.**

---

### Note · Authors · 2025-08-12

Dear Reviewers and AC, SAC, PC:

We sincerely thank all Reviewers and AC,SAC,PC for great effort and constructive comments on our manuscript.

As Reviewers highlighted, our paper **tackles an important problem (Reviewer pm1M,LRH2)**, and **introduces novel and brilliant ideas (Reviewer rQbD,9osN,LRH2,HrqG)**. We also appreciate that Reviewers found our paper **well-written (Reviewer pm1M,HrqG)** and **offers solid experiments (Reviewer 9osN,LRH2,HrqG)**.

Moreover, we **thank all reviewers for helpful questions and suggestions**. Fortunately, **we’ve fully addressed 4 Reviewers’ concerns and some doubts from Reviewer pm1M**. Grateful for their high scores and recognition, we **will provide more clarifications in the revised version.**

However, **Reviewer pm1M has some misunderstandings, which we have addressed comprehensively in our Rebuttal and Clarifications—regrettably without further feedback.** Below is a summary of key points:

* **(Q3) The scale of RCT data**: Has been revisited and clarified in our Rebuttal, and have received their recognition.
* **(Q1 and Q2) Reasons for the design of our Bi-DFCL**:
	1. **Our goal is to align prediction-decision objectives via $L\_{DL}$ while fully leveraging bias-variance complementary RCT and OBS data.**
	2. However, as emphasized in the main text, rebuttal, Clarification, **direct training of $L\_{DL}$ with RCT + OBS data is infeasible due to $L\_{DL}$’s extreme sensitivity to bias (see Clarification for details). This limitation is the core motivation of Bi-DFCL.**
	3. To address this limitation, we designed Bi-DFCL :

		- **Upper level**: Train $L\_{DL}$ exclusively on RCT data (consistent with existing works that design custom losses) to align with prediction-decision objectives
		- **Lower level**: Use massive OBS data (or OBS + RCT data, also compatible with our framework) to train parameterized $L\_{PL}(\phi,\theta)$, enhancing generalization and achieving optimal bias-variance tradeoff
		- **Bridge Network**: End-to-end trained to bridge the upper and lower level, dynamically correct  $L\_{PL}(\phi,\theta)$, render it both decision-aware and less biased
		- **Differentiation**: address two non-differentiability challenges via $ L\_{PPL}$ or $ L\_{PIFD}$ in Sec.4.2 and Implicit Differentiation in Sec.4.3
	4. **Overlooking our Bi-DFCL's motivation and focusing directly on design details may cause misunderstandings.**

Thanks again.

Authors of Paper 10859.

---

### Decision · Program_Chairs · 2025-09-17

**Decision:**

Accept (poster)

**Comment:**

The paper introduces a framework for addressing the prediction-decision misalignment in marketing optimization, bridging both observational and experimental data. The paper was well-received by most of the reviewers and inspired (what looks like) a fruitful exchange between the reviewers and the authors who addressed the reviewer concerns. One reviewer increased their score after the rebuttal. To meet the expectations of the reviewers, the authors should incorporate the rebuttal in their final paper, including improving the clarity of the paper, incorporating the computational cost results, and moving some of the sensitivity analysis to the main paper.